# UBE2O-mediated ubiquitylation directs cytoplasmic CTNNA1 to promote cell-to-ECM adhesions

Dan Xiang (ID) [1,2,3], Wenfeng Wu[4], Ruona Shi[5], Xiaoxiao Tang[1,2,3] & Xiaofei Zhang (ID) [1,2,3✉]

## Abstract

CTNNA1, a multifunctional protein that localizes at both the plasma membrane and the cytosol, plays crucial roles in actin dynamics regulation, cell-to-cell and cell-to-the extracellular matrix (ECM) adhesions and tumor suppression. Despite its diverse functions, the regulatory mechanisms by which cells coordinate CTNNA1's roles remain poorly understood. In this study, we identified UBE2O, a unique hybrid E2/E3 enzyme, as a key regulator that selectively interacts with and ubiquitylates cytosolic CTNNA1 in a phosphorylation-independent manner. Through comprehensive mass spectrometry-based interactome analysis of ubiquitylated CTNNA1, we reveal that the ubiquitylation of CTNNA1 diminishes its interaction with β-catenin while allowing its interaction with vinculin. This switch of molecular interactions promotes focal adhesions maturation, facilitates cell extension and matrix adhesion during the initial phases of cell spreading. Importantly, our findings demonstrate that ubiquitylation serves as a molecular switch that directs the regulatory roles of CTNNA1 to cell-to-ECM adhesions. This study advances our understanding of how ubiquitylation fine-tunes protein function in cell adhesion dynamics.

**Keywords** Cell Adhesions; CTNNA1; Proteomics; UBE2O; Ubiquitylation
**Subject Categories** Cell Adhesion, Polarity & Cytoskeleton; Post-translational Modifications & Proteolysis

## Introduction

Cell-to-cell and cell-to-the extracellular matrix (ECM) adhesions, along with their associated signalling pathways, are essential for the proper functioning of fundamental cell activities, such as adhesion, migration and proliferation (Iskratsch et al, 2014; Ringer et al, 2017). Cell-to-cell contacts are classically mediated by adherens junctions (AJs), which are characterized by a multiprotein complex comprising p120-catenin, β-catenin, and α-catenin. These components interact with the cytoplasmic tails of transmembrane cadherins, forming a relatively stable yet critical structure and signaling platform (Pokutta and Weis, 2007). In contrast, cell-to-ECM adhesions exhibit greater dynamism and continuity in their formation and turnover. These adhesion complexes are initiated by forming nascent adhesions, which then undergo rapid turnover or a maturation process: forming focal complexes and developing into elongated focal adhesions (FAs). FAs compose various proteins, including integrin, talin, zyxin and vinculin. In fibroblasts, FAs can further mature into fibrillar adhesions (Chastney et al, 2021; Jockusch et al, 1995; Parsons et al, 2010). The balance between these two adhesions plays a pivotal role in determining the cellular state and behaviours. However, the molecular mechanisms and regulatory pathways that govern the cellular decision-making processes in cell-to-cell or cell-to-ECM adhesions remain inadequately understood, representing a significant gap in our understanding of cellular adhesion regulation.

Among various cell adhesion-related proteins, the α-catenin protein has been conventionally recognized for its role in cell-to-cell adhesion through supporting AJs (Kobielak and Fuchs, 2004). A recent research, however, has suggested that it also plays a significant role in regulating cell-to-ECM adhesions through its interaction with vinculin (Mukherjee et al, 2022). Nevertheless, the precise mechanisms of how cells coordinate the involvement of α-catenin in cell-to-ECM adhesions remain to be fully elucidated. Notably, mammalian cells possess two distinct pools of α-catenin: one that is cytosolic and cadherin-free and another that is associated with cadherin-β-catenin complex at the cell membrane (Benjamin et al, 2010; Drees et al, 2005; Hinck et al, 1994; Schneider et al, 1993). The cytosolic pool of α-catenin plays a pivotal role in regulating actin dynamics by limiting Arp2/3-mediated actin polymerization (Benjamin et al, 2010; Drees et al, 2005; Hansen et al, 2013). Conversely, the membrane-associated α-catenin pool is essential for the formation and maintenance of AJs, as it links the cadherin-β-catenin complex and the actin cytoskeleton (Desai et al, 2013). In this process, α-catenin binds with β-catenin through its N-terminal domain and interacts with F-actin via its C-terminal domain. In addition, the M-domain (middle domain) of α-catenin recruits adhesion-related proteins, including F-actin-binding and integrins connection proteins such as α-actinin and vinculin (Ishiyama et al, 2013; Kobielak and Fuchs, 2004). However, the manner in which cells balance the regulatory

[1]Laboratory of Proteomics and Ubiquitin Signaling, Center for Cell Lineage Altas, Guangzhou Institutes of Biomedicine and Health, Chinese Academy of Sciences, 510530 Guangzhou, China. [2]Guangdong Provincial Key Laboratory of Stem Cell and Regenerative Medicine, Guangdong-Hong Kong Joint Laboratory for Stem Cell and Regenerative Medicine, Guangzhou Institutes of Biomedicine and Health, Chinese Academy of Sciences, 510530 Guangzhou, China. [3]University of Chinese Academy of Sciences, Beijing 100049, China. [4]Key Laboratory of Biological Targeting Diagnosis, Therapy and Rehabilitation of Guangdong Higher Education Institutes, the Fifth Affiliated Hospital of Guangzhou Medical University, 510799 Guangzhou, Guangdong, China. [5]Centre for Regenerative Medicine and Health, Hong Kong Institute of Science & Innovation, Chinese Academy of Sciences, 15 Science Park West Avenue, Hong Kong Science Park, Hong Kong SAR, China. ✉E-mail: zhang_xiaofei@gibh.ac.cn

roles of these functionally divergent α-catenin pools remains elusive.

In eukaryotes, the regulation of cellular processes is largely achieved through modular protein post-translational modifications (PTMs) (Dikic and Schulman, 2023). Within the PTMs repertoire, ubiquitylation represents a highly versatile mechanism that covalently attaches ubiquitin to substrate proteins (Clague et al, 2015). This cascade process is orchestrated by three types of enzymes: ubiquitin-activating enzymes (E1), ubiquitin-conjugating enzymes (E2), and ubiquitin ligases (E3) (Clague et al, 2015; Schulman and Harper, 2009; Uchida and Kitagawa, 2016; Ye and Rape, 2009). Of particular interest, UBE2O is one of only two identified hybrid E2/E3 enzymes capable of directly binding to substrates and facilitating the conjugation of ubiquitin to substrate proteins (Berleth and Pickart, 1996; Zhang et al, 2013a). UBE2O catalyzes ubiquitylation of diverse substrates, such as SMAD6, BAP1, RECQL4 and ribosomal proteins. These functions position UBE2O as a critical factor in orphan protein quality control, proteome remodeling, DNA repair, and cancer progression (Huang et al, 2022; Huang et al, 2021; Mashtalir et al, 2014; Nguyen et al, 2017; Ullah et al, 2019; Yanagitani et al, 2017; Zhang et al, 2013a). Despite these roles, the contribution of UBE2O to the modulation of cell adhesion processes remains poorly understood. Additionally, although it has been reported that the phosphorylation of α-catenin promotes intercellular adhesion (Escobar et al, 2015), the involvement of other PTMs, particularly ubiquitylation, in the regulation of α-catenin function in cell adhesions awaits characterization.

In human, there are three α-catenins: αE-catenin (CTNNA1), αN-catenin (CTNNA2) and αT-catenin (CTNNA3) (Kobielak and

Fuchs, 2004; Nagafuchi and Takeichi, 1989). In this study, we demonstrate that the cytoplasmic CTNNA1 selectively interacts with UBE2O and undergoes monoubiquitylation by UBE2O at multiple lysine residues. This modification facilitates CTNNA1 to preferentially interact with vinculin over β-catenin, thereby driving FAs maturation, facilitating cell extension and promoting cell-to-ECM adhesion during the early stages of cell spreading. Collectively, our results delineate a novel ubiquitylation-mediated signaling mechanism by which cytosolic CTNNA1 is directed to engage specifically in cell-to-ECM adhesion processes.

# Results

## UBE2O interacts with CTNNA1

To investigate the post-translational regulation of CTNNA1, we employed a mass spectrometry-based proteomic approach to study its interactome. Flag-tagged CTNNA1 was transiently overexpressed in HEK293T cells, allowing us to identify its interacting proteins through immunoprecipitation with Flag beads followed by mass spectrometry analysis. As illustrated in Fig. 1A and Dataset EV1, 180 proteins interacting with CTNNA1 were identified with high confidence (fold change >2 and $P < 0.05$), including known interactors such as CTNNB1 (β-catenin) and CDH1 (E-cadherin) (Benjamin et al, 2010; Drees et al, 2005; Koslov et al, 1997). Notably, several ubiquitin signaling-related proteins were also identified (Dataset EV1), including the E3 ubiquitin ligase RAD18, the deubiquitylase USP36, the E2/E3 hybrid enzyme

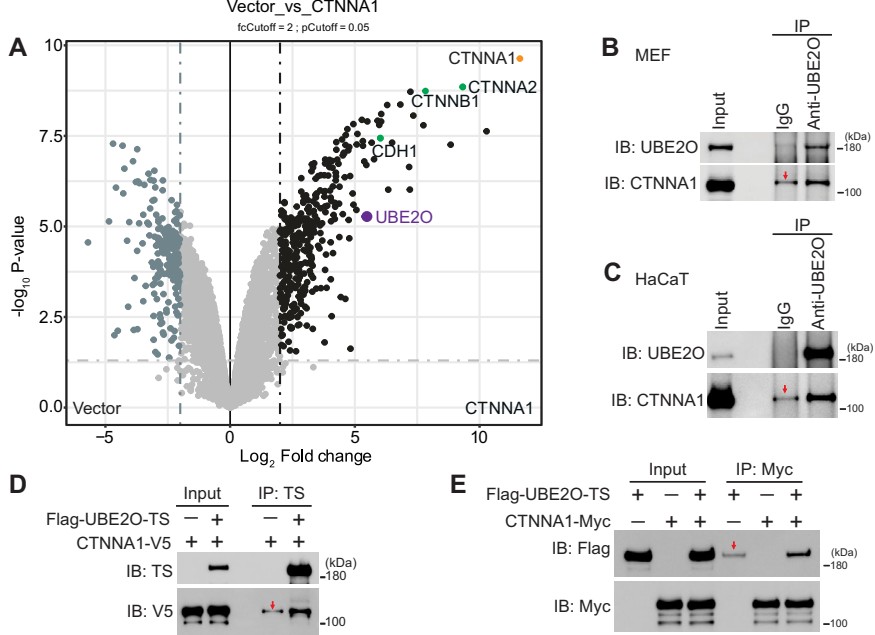

**Figure 1. Identification and validation of UBE2O as a CTNNA1-interacting partner.**

(A) Volcano plot displaying the significant interactors of CTNNA1 identified through mass spectrometry. Known interactors of CTNNA1 are highlighted in green dots for reference. Data were analysed and visualized using the DEP2 package, $n = 3$ technical replicates in independent experiments. Detailed data on all identified significant interactors are presented in Datasets EV1. (B–E) Validation of the interaction between UBE2O and CTNNA1 both in vivo (**B**, **C**) and in vitro (**D**, **E**). The red arrow indicates unspecific binding to the beads. All interaction experiments were repeated at least twice, and one representative result is shown. Source data are available online for this figure.

UBE2O and BIRC6, suggesting a potential role for ubiquitin signaling in the regulation of CTNNA1. Given that CTNNA1 was previously identified as a UBE2O interactor in an interactome screening effort (Nguyen et al, 2017), we decided to further investigate the interaction between CTNNA1 and UBE2O. We first validated the association between endogenously expressed UBE2O and CTNNA1 via co-immunoprecipitation using anti-UBE2O antibodies in mouse embryonic fibroblasts (MEFs) and the human keratinocyte cell line HaCaT cells (Fig. 1B,C). Additionally, reciprocal interactions were observed between purified UBE2O and CTNNA1 in vitro (Fig. 1D,E), suggesting that CTNNA1 directly interacts with UBE2O and may thereby serve as a potential substrate for UBE2O.

To delineate the specific domains of CTNNA1 and UBE2O that are responsible for their interaction, we constructed truncated mutations of both proteins. The truncated versions of CTNNA1 were generated according to its domain structure (Ishiyama et al, 2013) (Appendix Fig. S1A, top panel). The truncating of UBE2O was performed as previously described (Huang et al, 2022) (Appendix Fig. S1B, top panel). Co-expression of these truncated forms followed by co-immunoprecipitation analyses revealed that the N-terminal (N) domain of CTNNA1 and the conserved region 2 (CR2) domain of UBE2O are primarily responsible for their interaction (Appendix Fig. S1A,B, bottom panel). In summary, our results indicate that CTNNA1 directly interacts with UBE2O, positioning CTNNA1 as a potential substrate of UBE2O.

## UBE2O monoubiquitylates CTNNA1

Previous studies have reported that UBE2O ubiquitylates multiple proteins, such as RECQL4 and BAP1 (Huang et al, 2022; Mashtalir et al, 2014). To assess whether CTNNA1 was indeed a substrate of UBE2O, we performed a ubiquitylation assay under denaturing condition using 8 M urea buffer. As shown in Fig. 2A, UBE2O facilitated the ubiquitylation of CTNNA1 when co-expressed with ubiquitin. Importantly, we showed that the E2 active site, the cysteine residue at position 1040 within the UBC domain (Zhang et al, 2013a), is essential for UBE2O-mediated ubiquitylation of CTNNA1, as the catalytically inactive mutant UBE2O-C1040S (UBE2O-CS) was unable to ubiquitylate CTNNA1 (Fig. 2A). These findings were further validated using an alternative approach where proteins were denatured in a 1% SDS buffer, followed by a tenfold dilution for Flag-immunoprecipitation (Fig. 2B). Additionally, in vitro ubiquitylation reactions confirmed that UBE2O ubiquitylates CTNNA1 in its capacity as an E2/E3 hybrid enzyme (Appendix Fig. S2A). To corroborate the necessity of UBE2O for CTNNA1 ubiquitylation, we conducted knockdown experiments targeting endogenous *UBE2O* in HEK293T cells, resulting in a significant reduction of CTNNA1 ubiquitylation levels (Fig. 2C). Notably, two distinct bands of CTNNA1 were observed upon co-expression with UBE2O-D3 (Appendix Fig. S1B, lane 5). We proved that the upper band (indicated by a red asterisk) represents CTNNA1 modified by one ubiquitin molecule, as GlyGly(K) sites of CTNNA1 were identified after excising this band for in-gel digestion-based mass spectrometry analysis (Dataset EV1; Appendix Fig. S2B). Furthermore, we expressed UBE2O-D3 in HEK293T cells, denatured the proteins in cell lysates using a 1% SDS buffer, and subsequently performed a tenfold dilution for ubiquitin antibody (P4D1) immunoprecipitation to enrich ubiquitylated proteins. Consistent with our mass spectrometry data, we detected endogenous ubiquitylated CTNNA1 in cells overexpressing UBE2O-D3, but not in those overexpressing a control vector (Fig. 2D). Collectively,

these results demonstrate that UBE2O targets CTNNA1 for ubiquitylation both in vivo and in vitro.

We observed multiple UBE2O-mediated ubiquitylation bands for CTNNA1 (Fig. 2A–C; Appendix Fig. S2A), leading us to speculate that UBE2O may facilitate either polyubiquitylation or multi-monoubiquitylation of CTNNA1. To investigate this further, we performed an in vitro ubiquitylation assay using a lysine-deficient (KO) mutant of ubiquitin, which cannot form any polyubiquitylation linkage. As shown in Fig. 2E, multiple ubiquitylation bands for CTNNA1 remained detectable when reacted with the KO ubiquitin, indicating that UBE2O mediates multi-monoubiquitylation of CTNNA1 in vitro.

To pinpoint the specific lysine residues in CTNNA1 that could be subject to multi-monoubiquitylation, we co-expressed UBE2O, CTNNA1, and ubiquitin in Expi293F cells, followed by a two-step enrichment of ubiquitylated CTNNA1 and in-gel digestion-based proteomic identification (Huang et al, 2022). Compared to samples co-expressing UBE2O-CS, we observed a marked increase in the intensities of ubiquitylated peptides corresponding to K45, K81, K155 and K747 sites when co-expressed with UBE2O wild-type (WT), indicating these lysine residues as potential ubiquitylation sites mediated by UBE2O (Dataset EV1; Appendix Fig. S2C). However, assays employing single or combined mutations (5KR) of these identified lysine residues did not significantly reduce UBE2O-mediated ubiquitylation of CTNNA1 (Appendix Fig. S2D). This phenomenon aligns with the "alternative ubiquitylation theory", which proposes that when canonical ubiquitylation sites are inaccessible, compensatory ubiquitylation may occur at other lysine residues of the substrate (Dimova et al, 2012; Li et al, 2014). Moreover, this phenomenon was previously observed with RECQL4 ubiquitylation mediated by UBE2O (Huang et al, 2022). Alternatively, we mutated all lysine residues in CTNNA1 to arginine residues (CTNNA1-KO), resulting in the complete abolition of UBE2O-mediated CTNNA1 ubiquitylation (Fig. 2F; Appendix Fig. S2E). Additionally, we explored whether UBE2O could also ubiquitylate CTNNA2, another isoform of α-catenin. Interaction and ubiquitylation assays revealed that UBE2O interacted with both CTNNA1 and CTNNA2 (Appendix Fig. S2F), but only ubiquitylated CTNNA1 (Appendix Fig. S2G). In conclusion, these findings establish that UBE2O specifically facilitates monoubiquitylation of CTNNA1 at lysine residues.

## Ubiquitylation of CTNNA1 mediated by UBE2O is independent of its S641 phosphorylation

Given the established significance of phosphorylation at serine 641 (S641) of CTNNA1 in various cellular processes (Escobar et al, 2015; Ji et al, 2009; Shen et al, 2022), and the known interplay between phosphorylation and ubiquitylation in another catenin (Wu et al, 2020), we aimed to investigate whether the phosphorylation state of CTNNA1 influenced its ubiquitylation. To explore this, we expressed Myc-tagged CTNNA1 in HEK293T cells and subsequently enriched it using anti-c-Myc magnetic beads. Following this, we treated the enriched CTNNA1 with Lambda Protein Phosphatase (Lambda PP) to remove phosphate groups. After eliminating Lambda PP through washing, we added purified mUBE1, ubiquitin, and UBE2O to conduct an in vitro ubiquitylation assay to determine if CTNNA1 could still be ubiquitylated by UBE2O after dephosphorylation (Fig. 3A). As illustrated in Fig. 3B,

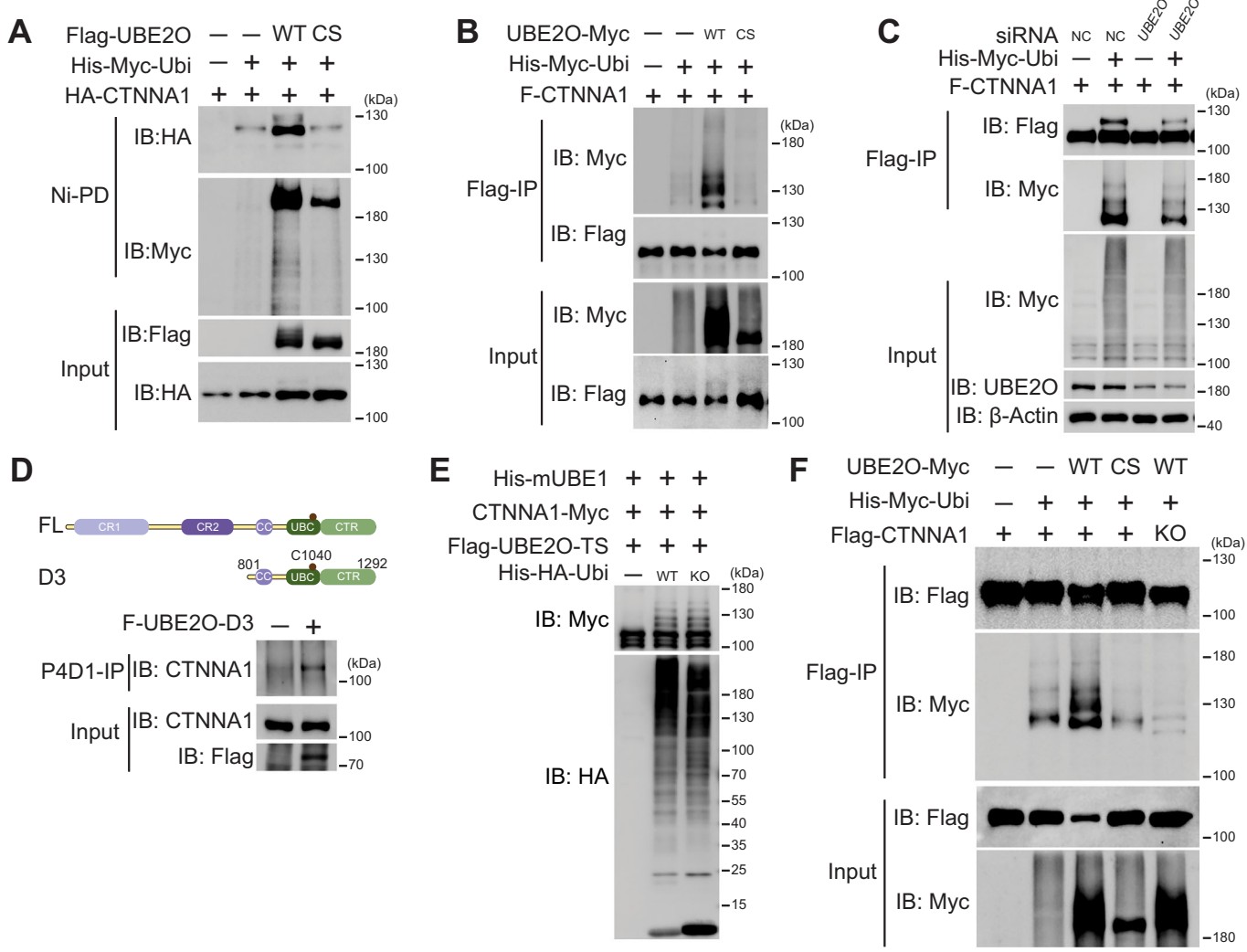

**Figure 2. UBE2O monoubiquitylates CTNNA1.**

(A, B) Immunoblots show UBE2O promotes CTNNA1 ubiquitylation and its E2 catalytic site is required for the ubiquitylation. His-Myc-Ubi, 6 × His-Myc-ubiquitin, WT wild-type, CS C1040S mutation. (C) Immunoblot shows UBE2O depletion reduces CTNNA1 ubiquitylation. NC negative control. (D) Immunoblot shows that UBE2O-D3 promotes the ubiquitylation of endogenous CTNNA1. (E) In vitro ubiquitylation assay shows UBE2O multi-monoubiquitylates CTNNA1 in vitro. KO lysine-deficient mutant. (F) Immunoblot shows UBE2O ubiquitylates CTNNA1 at multiple lysine residues. All experiments were repeated at least twice, one representative result is shown. Source data are available online for this figure.

treating with Lambda PP effectively reduced the phosphorylation of CTNNA1 at S641 but did not significantly impact the UBE2O-mediated ubiquitylation of CTNNA1. Additionally, we assessed the ubiquitylation potential of both the S641A (SA) mutant, which is designed to prevent phosphorylation by substituting serine with alanine, and the phosphorylation-mimetic S641D (SD) mutant, where serine is replaced with aspartic acid. Both mutants were capable of undergoing ubiquitylation mediated by UBE2O (Fig. 3C), reinforcing the conclusion that UBE2O-mediated ubiquitylation of CTNNA1 is independent of its phosphorylation at S641.

We were also interested in determining whether the ubiquitylation of CTNNA1 was necessary for its phosphorylation at S641. To investigate this, we ubiquitylated CTNNA1 in HEK293T cells by co-transfecting UBE2O-D3, ubiquitin, and CTNNA1. We observed that the CTNNA1 was phosphorylated irrespective of the presence or absence of UBE2O-D3 (Fig. 3D, lanes 1–3). Notably, the lysine-deficient

KO mutant, which abolished UBE2O-mediated ubiquitylation (Figs. 2F and 3D; Appendix Fig. S2E), still exhibited phosphorylation at S641 (Fig. 3D, lanes 4–6) compared with the SA mutant. Furthermore, we observed that both phosphorylation and ubiquitylation can coexist on CTNNA1 (Fig. 3D, lanes 1–3). These findings collectively suggest that the ubiquitylation and S641 phosphorylation of CTNNA1 are independent post-translational modifications, each occurring without influence from the other.

## E2-conjugating activity of UBE2O is required for FAs maturation and cell extension

A previous study has indicated that the knockdown of *Ctnna1* leads to a reduction in mature FAs and a consequent decrease in cell area at the early stages of cell spreading (Mukherjee et al, 2022), which were identical to our results (Figs. EV1A and EV5A,B). To explore

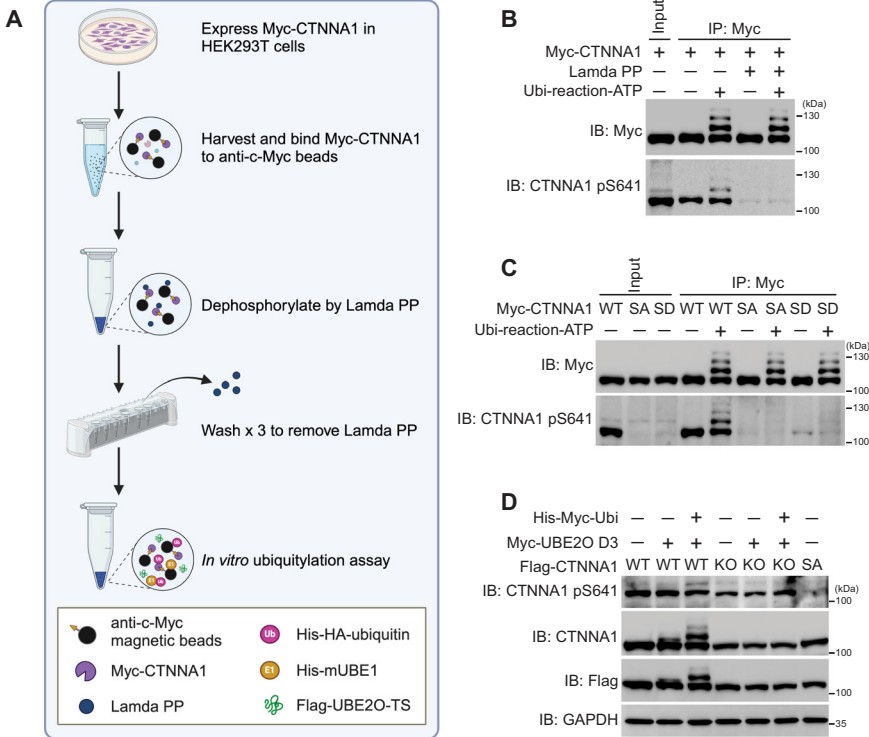

**Figure 3. Ubiquitylation of CTNNA1 mediated by UBE2O is independent of its S641 phosphorylation.**

(A) Schematic diagram showing the assay of dephosphorylation followed by ubiquitylation. Detailed description is shown in the method. This diagram was created in BioRender. (B, C) Phosphorylation is not a prerequisite for UBE2O-mediated CTNNA1 ubiquitylation. (B) CTNNA1-Myc protein expressed in HEK293T cells was enriched for dephosphorylation by Lambda PP. Following this, an in vitro ubiquitylation reaction was performed, with samples incubated with ATP (Ubi-reaction ATP) or without ATP. (C) CTNNA1-Myc or its S641A (SA) and S641D (SD) mutants expressed in HEK293T cells were enriched to anti-c-Myc beads to perform in vitro ubiquitylation reaction with ATP (Ubi-reaction ATP) or without ATP. (D) Phosphorylation of CTNNA1 at S641 is independent of UBE2O-mediated ubiquitylation. HEK293T cells were transfected with the indicated plasmids for 36 h. Cells were harvested for western blot analyses. All experiments were repeated at least twice, one representative result is shown. Source data are available online for this figure.

the potential role of UBE2O in the regulation of cell adhesion to the ECM, we sought to investigate the localization of UBE2O at cell-to-ECM adhesions. Because none of the commercial mouse-originated antibodies worked well to detect the endogenous UBE2O and zyxin (a known marker for mature FAs) during the immunostaining assay, we alternatively constructed MEFs that stably express GFP-tagged zyxin to facilitate the observation of FAs (Fig. EV1B). Interestingly, via Lattice Structured Illumination Microscopy (SIM), we observed that at the early stages of cell spreading, UBE2O presented at the cell edge and co-localized with CTNNA1 which displayed striped patterns at zyxin-containing FAs (Fig. 4A). Specifically, both UBE2O and CTNNA1 proteins are detectable on small FAs, with their presence becoming more prominent in mature FAs and larger fibrillar adhesions (Fig. 4A). Furthermore, knockdown of Ube2o using two different shRNAs, targeting either the 3'-UTR or the coding sequence (CDS), resulted in cells developing predominantly small adhesions at the cell edge, along with a notable decrease in cell area (Fig. EV1C,D). These findings support our speculation that UBE2O plays a regulatory role in FAs. Moreover, further experiments demonstrated that the expression of Flag-UBE2O-WT in Ube2o knockdown MEFs could restore the maturation of FAs, whereas the expression of the catalytically inactive Flag-UBE2O-CS mutant failed to achieve this restoration (Figs. 4B and EV1E). Corroborating their impaired ability in

adhesions maturation, both Ube2o knockdown cells and cells expressing Flag-UBE2O-CS mutant displayed significantly reduced cell sizes compared to control cells or cells expressing Flag-UBE2O-WT (Figs. 4B and EV1E). Similar phenotypes were observed in Madin-Darby canine kidney (MDCK) cells (Figs. 4C and EV1F). In conclusion, these results support that the E2-conjugating activity of UBE2O is essential for the regulatory role of UBE2O in FAs maturation and cell extension during the initial phases of cell spreading.

## UBE2O-mediated ubiquitylation of CTNNA1 alters its interactome and subcellular localization

To elucidate the underlying mechanism by which UBE2O regulates FAs maturation, we first investigated whether UBE2O affects the maturation of adhesions by governing the protein levels of CTNNA1 via monoubiquitylation. Previous research has indicated that UBE2O-mediated multi-monoubiquitylation can impact the stability and degradation of substrate proteins, such as RECQL4 (Huang et al, 2022). However, our findings revealed that knockdown of Ube2o in both MEFs and MDCK cells did not lead to significant changes in CTNNA1 protein level (Fig. EV2A,B). Correspondingly, re-introducing UBE2O-WT, the catalytically inactive UBE2O-CS mutant, or a control vector had no considerable

effect on CTNNA1 protein level (Fig. EV2C,D). Furthermore, a gradual knockdown of *UBE2O* in U-2 OS cells using doxycycline-induced shRNAs also did not result in substantial changes in CTNNA1 protein level (Fig. EV2E). Additionally, ectopic expression of UBE2O in U-2 OS cells did not affect CTNNA1 protein level (Fig. EV2F). These results are consistent with an earlier study suggesting that CTNNA1 is degraded by a mechanism that is independent of ubiquitylation (Hwang et al, 2005).

Beyond influencing degradation, monoubiquitylation can alter the functions of substrate proteins in various ways, including changing subcellular localization, impacting trafficking, regulating activity, or affecting protein-protein interactions (Huang and Zhang, 2020). Given that the ubiquitylation sites identified on CTNNA1 are located in functionally critical regions that interact with proteins such as β-catenin, actin, and vinculin (Kobielak and Fuchs, 2004), we aimed to investigate the interactome of CTNNA1

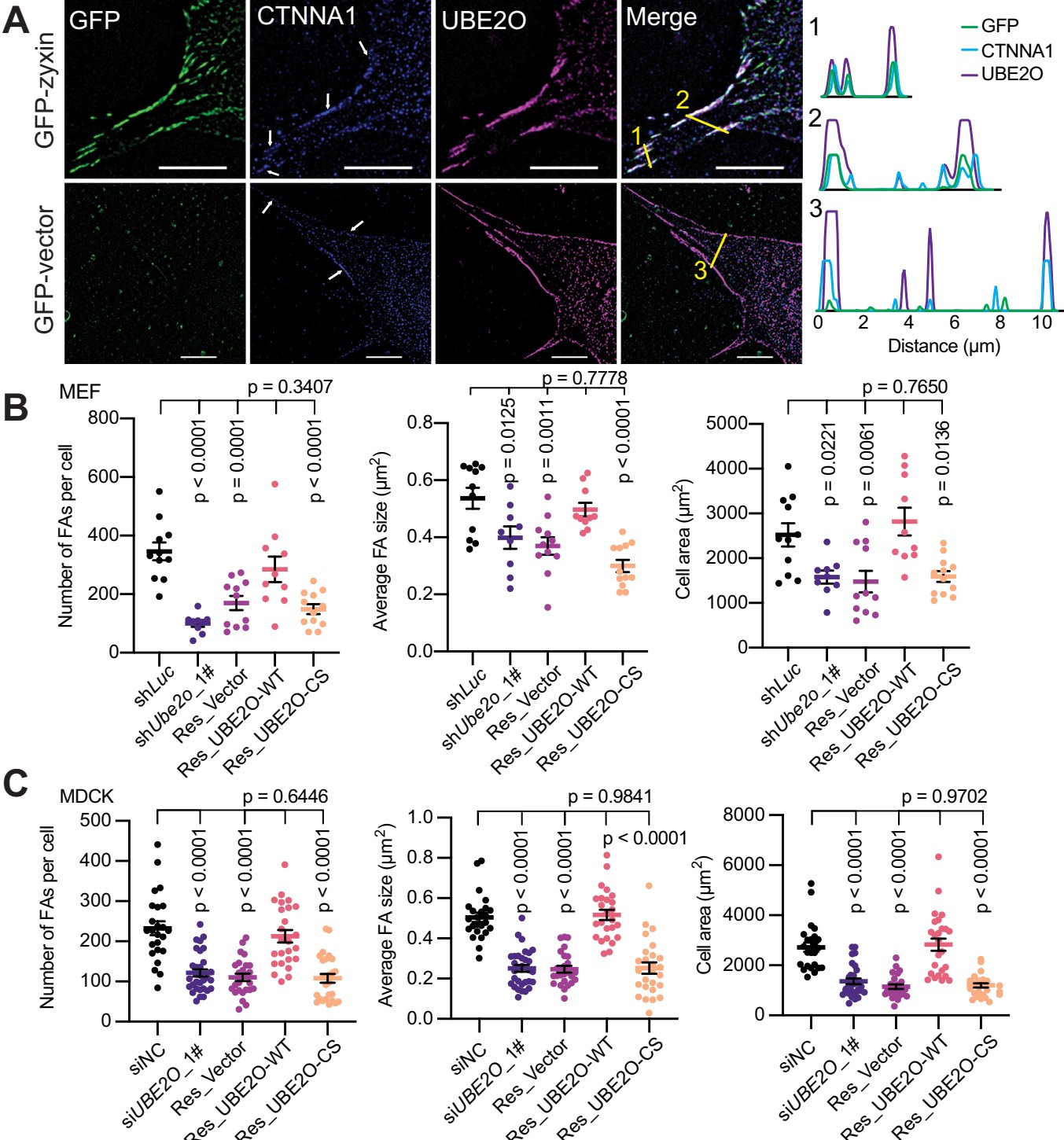

◄  **Figure 4.   E2-conjugating activity of UBE2O is required for FAs maturation and cell extension.**

(A) Representative micrographs show that UBE2O colocalizes with CTNNA1 at zyxin-containing FAs. The graphs on the right are the gray values of UBE2O (purple), CTNNA1 (blue) and GFP (green) measured along the yellow lines in the merged image. White arrows point to striped locations of CTNNA1 at the cell edge. Scale bar: 10 μm. (B) The catalytic activity of UBE2O is required for FAs maturation and cell extension in MEFs. Quantifications of the number of FAs per cell (zyxin channel), average FA size (zyxin channel) and cell area (phalloidin channel) of the indicated MEFs 6 h after plating on FN-coated 8-cell chamber slides (representative micrographs are shown in Fig. EV1E). Total number of cells analyzed: shLuc (11), shUbe2o_1# (9), Res_Vector (11), Res_UBE2O-WT (10), Res_UBE2O-CS (12). (C) UBE2O is responsible for FAs maturation in MDCK cells and is catalytic activity-dependent. Quantifications of the number of FAs per cell (zyxin channel), average FA size (zyxin channel) and cell area (zyxin channel) of the indicated MDCK cells 8 h after plating on 8-cell chamber slides (representative micrographs are shown in Fig. EV1F). Total number of cells analyzed: siNC (24), siUBE2O_2# (31), Res_Vec (25), Res_UBE2O-WT (25), Res_UBE2O-CS (26). Error bars indicate mean ± SEM. Significance among multiple groups was determined using ANOVA followed by Tukey's post hoc test. Experiments were repeated at least twice, one representative result is shown. Source data are available online for this figure.

upon ubiquitylation. To accomplish this, we developed a strategy to prepare ubiquitylated CTNNA1 and identify its interactors (Fig. EV3A). Specifically, we catalyzed the ubiquitylation of CTNNA1-V5 protein using UBE2O-D3 in an in vitro ubiquitylation reaction, followed by incubating the reaction supernatants with V5-Trap magnetic agarose to enrich the ubiquitylated CTNNA1 (CTNNA1$_{Ub}$) bait. We included two control groups: one without CTNNA1-V5 (ctrl) and another without ATP (CTNNA1) in the ubiquitylation reaction, to serve as comparisons. After incubating these baits (Fig. 5A) with cell lysates, we employed on-bead digestion and mass spectrometry to identify the interactome. The mass spectrometry data indicated that ubiquitylation altered the interactions between CTNNA1 and various proteins (Fig. 5B): some interactions were reduced (Cluster 1) while others were increased (Cluster 2). We conducted Gene Ontology (GO) analysis on proteins within Cluster 1 and found significant enrichment in multiple biological processes associated with cell adhesion, suggesting that ubiquitylation affects the interaction of CTNNA1 with proteins integral to cell adhesion complexes (Fig. 5C). We selected the proteins with decreased interaction, CTNNA1, CTNNA2, and β-catenin, alongside the increased interaction protein vinculin (Fig. EV3B), to validate the mass spectrometry findings through immunoprecipitation and immunoblotting experiments in HEK293T cells (Figs. 5D and EV3C), MDCK cells (Fig. 5E), and MEFs (Fig. EV3D). Moreover, we found that in Ube2o knockdown MEFs, endogenous CTNNA1 showed increased interaction with β-catenin while decreased binding to vinculin compared to controls (Fig. EV3E). These results confirmed that the ubiquitylation of CTNNA1 influences its binding ability to various adhesion-related proteins.

To explore further, we investigated the influence of UBE2O on the localization of CTNNA1 to FAs, as UBE2O-mediated ubiquitylation of CTNNA1 alters its binding ability to various adhesion-related proteins. As shown in Fig. 5F, CTNNA1 displayed striped patterns at the cell edge that coincided with FAs, especially mature FAs, as indicated by zyxin in shLuc MEFs. However, knockdown of Ube2o resulted in cells primarily developing small, nascent adhesions at the cell edge, with decreased CTNNA1 localization to these structures. Moreover, the expression of UBE2O-WT, but not the catalytically inactive UBE2O-CS mutant, in the Ube2o knockdown MEFs restored both the maturation of FAs and CTNNA1's localization at these sites (Fig. 5F). In conclusion, these results indicate that the regulatory role of UBE2O in cell adhesion may be achieved by altering the interactome of ubiquitylated CTNNA1, thereby affecting its subcellular localization at FAs.

## UBE2O-mediated ubiquitylation of CTNNA1 is required for FAs maturation and cell extension

To investigate whether and how the changes in the interactome and subsequent subcellular localization of CTNNA1 contributed to UBE2O's role in FAs maturation, we focused on the protein interactions that changed significantly post-ubiquitylation: decreased interaction with β-catenin and increased interaction with vinculin. Previous studies have highlighted the critical role of the CTNNA1-vinculin interaction in FAs maturation (Mukherjee et al, 2022) and CTNNA1-β-catenin interaction in the formation and maintenance of AJs (Kobielak and Fuchs, 2004). Therefore, we hypothesized that UBE2O-mediated ubiquitylation of CTNNA1 facilitates its engagement with vinculin at the expense of its interaction with β-catenin, thereby promoting CTNNA1's involvement in cell-to-ECM adhesions as opposed to cell-to-cell adhesions. To validate this hypothesis, we first stained Ube2o knockdown and control MEFs with vinculin. As illustrated in Fig. EV4A,B, knockdown of Ube2o resulted in the formation of smaller vinculin-containing FAs accompanied by a reduction in cell spreading. However, the expression of UBE2O-WT, but not the catalytically inactive UBE2O-CS mutant, in the Ube2o knockdown MEFs restored both the maturation of vinculin-containing FAs and cell area (Figs. 6A and EV4C). These findings confirm that the catalytic activity of UBE2O is essential for the maturation of vinculin-containing FAs.

Next, we investigated the role of CTNNA1 ubiquitylation in UBE2O-mediated adhesion maturation by employing CTNNA1-KO, a ubiquitylation-resistant mutant (Figs. 2F and 3D; Appendix Fig. S2E). The knockdown of Ctnna1 significantly impaired FAs maturation and led to a marked decrease in cell area (Figs. 6B and EV4D). Reintroduction of CTNNA1-WT restored the maturation of adhesions, whereas the KO mutant failed to achieve this restoration (Figs. 6B and EV4D). Immunoprecipitation assays in MEFs stably expressing Flag-tagged CTNNA1-KO confirmed its interaction with endogenous β-catenin (Fig. EV4E), demonstrating that the mutant protein retains native folding and binding capacity. This result excludes protein misfolding as a potential cause for the negative rescue phenotype observed with CTNNA1-KO. Additionally, knockdown of Ctnna1 in MEFs with UBE2O overexpression abolished the adhesion maturation driven by UBE2O (Figs. 6C and EV4F), indicating that the ubiquitylation of CTNNA1 is critically required for UBE2O-mediated adhesion maturation. Furthermore, to rule out the possibility that the disability in vinculin-containing FAs maturation was due to a deficiency in vinculin recruitment to these adhesions, we stained these cells with another FA marker zyxin and confirmed a significant decrease in the number and average size of adhesions

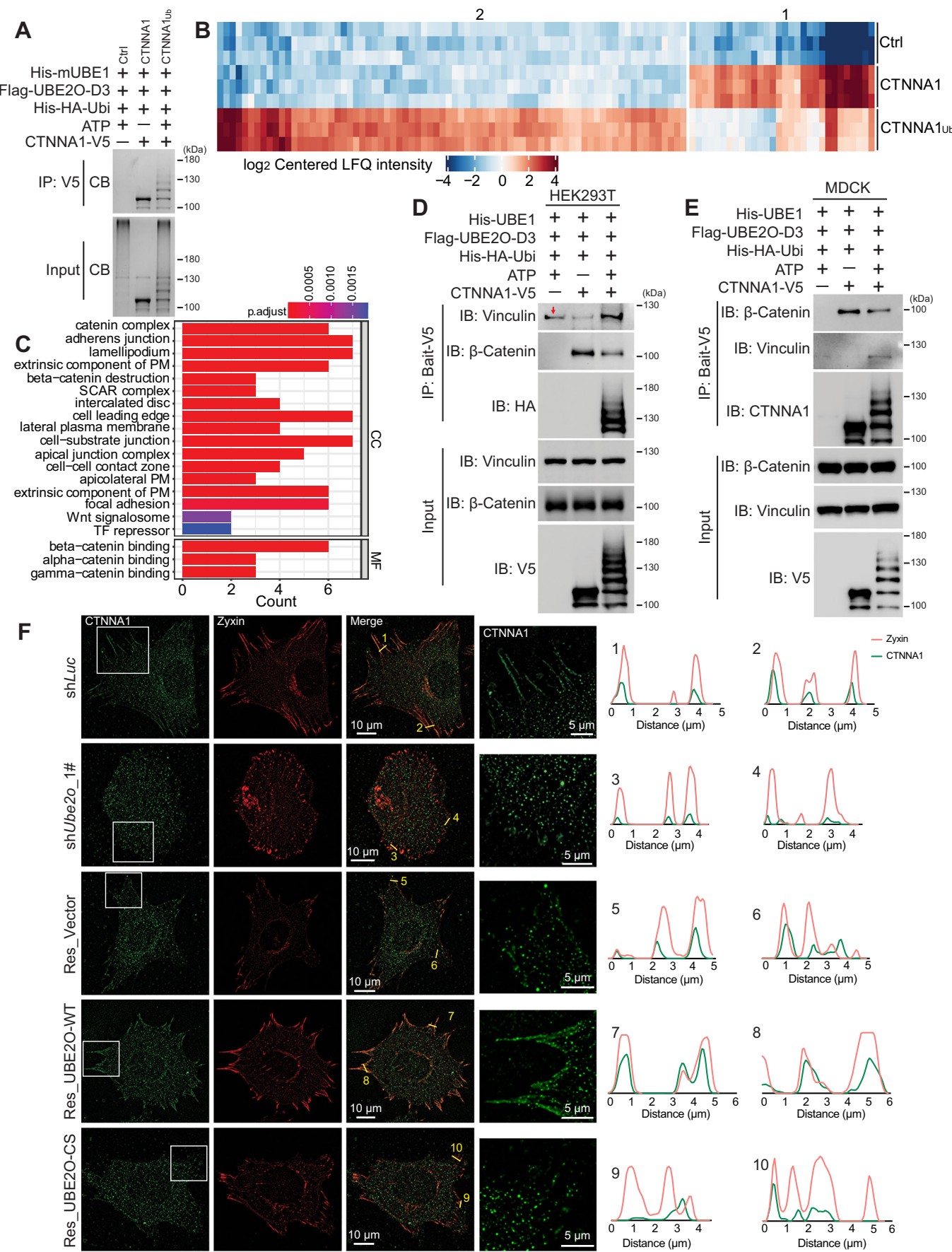

◀

**Figure 5. UBE2O-mediated ubiquitylation of CTNNA1 alters its interactome and subcellular localization.**

(A) Coomassie blue staining of the baits made for ubiquitylated CTNNA1 (CTNNA1$_{Ub}$) interactors identification. In vitro ubiquitylation reactions as indicated were carried out at 37 °C for 1 h. A portion of the reaction supernatants was collected as input, while the remaining supernatants were subsequently incubated with V5-trap beads for 2 h. The baits bound to beads were then eluted and identified by Coomassie blue staining (CB). (B) Heatmap of ubiquitylated CTNNA1 (CTNNA1$_{Ub}$) interactome. Data were analyzed and visualized by DEP2 package, $n = 3$ technical replicates in independent experiments. (C) Gene Ontology (GO) analysis of Cluster 1 proteins in Fig. 5B. Data were analyzed and visualized by DEP2 package. (D, E) Validation of the interaction between ubiquitylated CTNNA1 and vinculin or β-catenin by immunoprecipitation coupled with immunoblotting experiments in HEK293T (D) and MDCK (E) cells. The band indicated by the red arrow indicates unspecific binding to magnetic beads. (F) E2-conjugating activity of UBE2O contributes to CTNNA1's subcellular localization at FAs. The indicated MEFs co-stained for CTNNA1 and zyxin after 6 h of spreading on an FN-coated 8-cell chamber slide; the right panel is a zoom-in of the box in the left image, showing CTNNA1 strips at the cell edge; the graphs on the right are the gray values of zyxin (red) and CTNNA1 (green) measured along the yellow lines in the merged image. Interaction and immunofluorescence experiments were repeated at least twice, one representative result is shown. Source data are available online for this figure.

(Figs. 4B,C, EV1E,F and EV5A–D). Collectively, these results substantiate the conclusion that UBE2O-mediated ubiquitylation of CTNNA1 effectively drives CTNNA1's interactions toward vinculin to mediate FAs maturation and cell extension at the early stages of cell spreading.

## UBE2O selectively engages cytosolic CTNNA1 in FAs rather than AJs

It is established that cells possess two pools of α-catenin, cytoplasmic and membrane-associated fractions (Benjamin et al, 2010; Drees et al, 2005; Hinck et al, 1994; Schneider et al, 1993). Supported by our data, we speculated that UBE2O preferentially ubiquitylates cytosolic free CTNNA1 rather than the pool of CTNNA1 associated with cadherin-β-catenin complex. It is established that CTNNA1 binds to β-catenin via its N-terminal domain, and to F-actin through its C-terminal domain (Kobielak and Fuchs, 2004). Because UBE2O also interacts with these domains of CTNNA1 (Appendix Fig. S1A), β-catenin probably competes against UBE2O for binding to CTNNA1. To investigate this hypothesis, we conducted an experiment wherein we pre-incubated purified CTNNA1-V5 protein with or without purified His-GST-β-catenin. Following this, we used V5-trap beads to enrich for either CTNNA1 or the CTNNA1-β-catenin complex, subsequently adding purified Flag-UBE2O-TwinStrep proteins to assess their interaction in vitro. The results demonstrated that pre-binding CTNNA1 with β-catenin effectively inhibited its interaction with UBE2O in a dose-dependent manner (Fig. 7A). Consistently, co-expression with β-catenin reduced the interaction of UBE2O and CTNNA1 in HEK293T cells (Appendix Fig. S3A). Furthermore, the ubiquitylation of CTNNA1 mediated by UBE2O was decreased when β-catenin was co-transfected in a dose-dependent manner (Fig. 7B). To make a more profound investigation, we transfected UBE2O-D3 alongside CTNNA1 and ubiquitin to induce CTNNA1 ubiquitylation in cells and performed subcellular fractionation analysis to ascertain which pool of CTNNA1 was ubiquitylated. As illustrated in Fig. 7C and Appendix Fig. S3B, the cytosolic and cadherin-free pool of CTNNA1 was the fraction that was ubiquitylated by UBE2O-D3 both in HEK293T and MDCK cells. Collectively, these results suggest that UBE2O selectively interacts with and ubiquitylates cytosolic CTNNA1.

Based on our findings that UBE2O selectively ubiquitylates cytosolic CTNNA1, it can be theorized that any changes in the interaction between cytosolic ubiquitylated CTNNA1 and vinculin or β-catenin would not significantly strengthen or disrupt AJs. Furthermore, our results indicate that the phosphorylation of CTNNA1, which has been reported to enhance the strength of cell-

to-cell adhesions (Escobar et al, 2015), is independent of the ubiquitylation of CTNNA1 (Fig. 3A–D). This absence of cross-talk further supports the notion that ubiquitylated CTNNA1 may not participate in cell-to-cell adhesions. These speculations are supported by the immunofluorescence results obtained in UBE2O-overexpressing MDCK cells (Appendix Fig. S3C). To further validate this, we employed MDCK cells at early stages of spreading, during which both cell-to-ECM and cell-to-cell adhesions are active and there is a shift in the balance between both adhesion types. As illustrated in Fig. 7D, knockdown of UBE2O led to the formation of smaller FAs without significant changes in AJs formation. Furthermore, the expression of UBE2O-WT, but not the catalytically inactive UBE2O-CS mutant, in the UBE2O knockdown MDCK cells restored the maturation of FAs without affecting AJs. These findings confirm that UBE2O selectively directs CTNNA1 to be involved in FAs rather than AJs.

## UBE2O-CTNNA1 axis is required for cell adhesion to the ECM

Beyond its role in adhesion regulation, CTNNA1 has been identified as a tumor suppressor (Silvis et al, 2011; Sun et al, 2014). In contrast, UBE2O has been associated with both tumor-promoting and tumor-inhibiting functions (Vila et al, 2017; Xu et al, 2017). Since FAs maturation mediated by UBE2O-catalyzed CTNNA1 ubiquitylation could hypothetically reduce cell migration, it is possible that the tumor-inhibiting properties of UBE2O are linked to this function. However, transwell assays conducted in UBE2O-overexpressing MEFs following Ctnna1 knockdown showed no significant migration-promoting effects (Appendix Fig. S4A). This may be because the UBE2O-CTNNA1 axis functions during the early stages of cell spreading, which may not be sufficient to influence long-term cell migration, or its effects may be context-dependent. Next, we investigated the influence of UBE2O-CTNNA1 axis on cell adhesion to the ECM. As shown in Fig. 8A, knockdown of Ube2o in MEFs led to a decreased adhesion rate to the ECM. Notably, the expression of UBE2O-WT, but not the control vector or catalytically inactive UBE2O-CS mutant, in the Ube2o knockdown MEFs restored the adhesion rate to the ECM. Interestingly, the UBE2O-enhanced adhesion rate to the ECM was strongly inhibited when Ctnna1 was knocked down in MEFs (Fig. 8B). These findings confirm that UBE2O-CTNNA1 axis is required for cell-to-ECM adhesions during the early stages of cell spreading.

In conclusion, our study finds that UBE2O selectively catalyzes the ubiquitylation of cytosolic CTNNA1, which then changes CTNNA1's preference toward β-catenin to vinculin to impact FAs but not AJs. As a result, these processes facilitate cell extension and

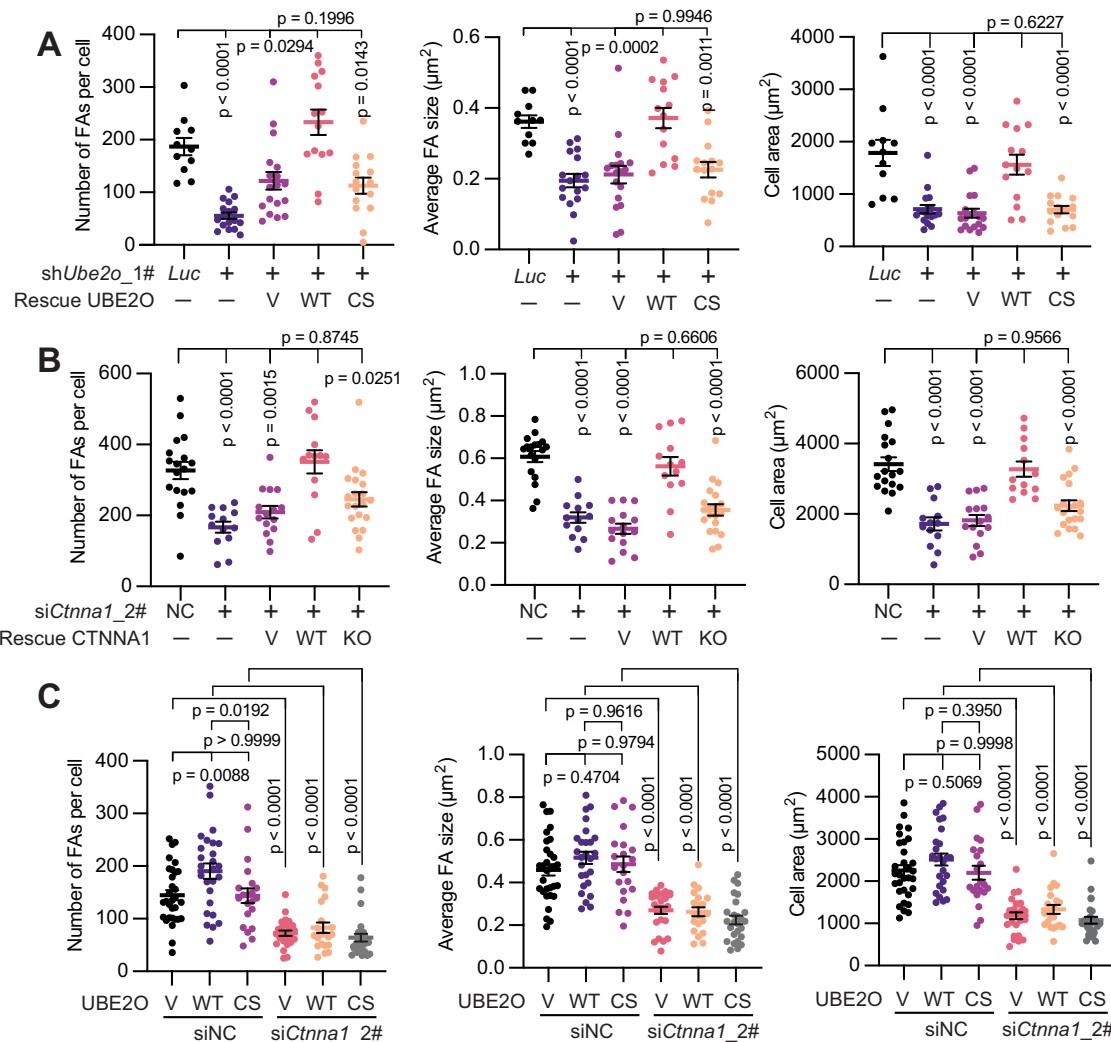

**Figure 6. UBE2O-mediated ubiquitylation of CTNNA1 is required for FAs maturation and cell extension.**

(A) The catalytic activity of UBE2O is responsible for vinculin-containing FAs maturation. Quantifications of the number of FAs per cell (vinculin channel), average FA size (vinculin channel) and cell area (vinculin channel) of the indicated MEFs (representative micrographs are shown in Fig. EV4C) 6 h after plating on FN-coated 8-cell chamber slides. Total number of cells analyzed: shLuc (11), shUbe2o_1# (17), Res_Vector (V) (18), Res_UBE2O-WT (14), Res_UBE2O-CS (15). (B) Ubiquitylation of CTNNA1 is engaged in FAs maturation and cell extension. Quantifications of the number of FAs per cell (vinculin channel), average FA size (vinculin channel) and cell area (phalloidin channel) of the indicated MEFs 6 h after plating on FN-coated 8-cell chamber slides (representative micrographs are shown in Fig. EV4D). Total number of cells analyzed: siNC (18), siCtnna1_2# (13), Res_Vector (V) (15), Res_CTNNA1-WT (13), Res_CTNNA1-KO (20). (C) UBE2O regulates FAs maturation via CTNNA1. Quantifications of the number of FAs per cell (vinculin channel), average FA size (vinculin channel) and cell area (vinculin channel) of the indicated MEFs 6 h after plating on FN-coated 8-cell chamber slides (representative micrographs are shown in Fig. EV4F). Total number of cells analyzed: Vector + siNC (31), UBE2O-WT + siNC (26), UBE2O-CS + siNC (21), Vector + siCtnna1_2# (26), UBE2O-WT + siCtnna1_2# (21), UBE2O-CS + siCtnna1_2# (26). Error bars indicate mean ± SEM. Significance among multiple groups was determined using ANOVA followed by Tukey's post hoc test. All experiments were repeated at least twice, one representative result is shown. Source data are available online for this figure.

adhesion to the ECM during the early stages of cell spreading, therefore preferentially directing CTNNA1 to cell-to-ECM adhesions rather than cell-to-cell adhesions (Fig. 8C).

## Discussion

α-Catenin plays a critical role in regulating both cell-to-cell and cell-to-ECM adhesions (Desai et al, 2013; Drees et al, 2005; Ishiyama et al, 2013; Mukherjee et al, 2022). In order to discern the

mechanisms by which CTNNA1 is regulated and selectively engaged in these different adhesion types, we investigated its PTM and identified UBE2O as the enzyme to mediate the ubiquitylation of CTNNA1. By combining in vitro reconstitution of ubiquitylated CTNNA1 with advanced mass spectrometry-based proteomics, we generated the first interactome map of ubiquitylated CTNNA1. This analysis revealed that ubiquitylation has led to significant changes in the interaction of CTNNA1 with numerous proteins, many of which are closely related to cell adhesion processes. As a mechanosensory protein, CTNNA1 typically binds

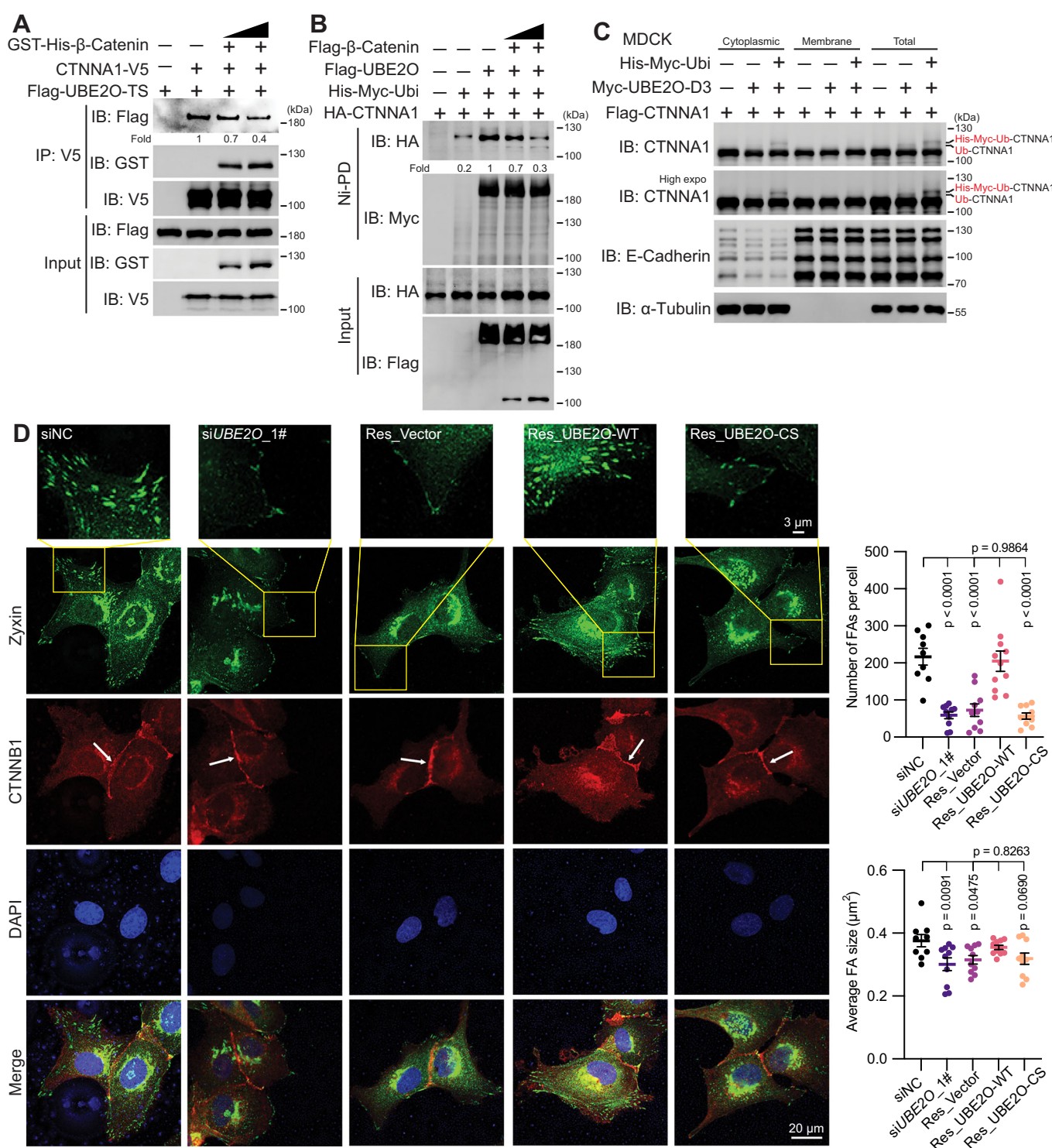

to vinculin when mechanical forces unfold its auto-inhibited M-domain, rendering the vinculin-binding site accessible (Buckley et al, 2014; Thomas et al, 2013; Yao et al, 2014; Yonemura et al, 2010). Intriguingly, our study demonstrates that ubiquitylation increases the interaction of CTNNA1 with vinculin, introducing an alternative regulatory mechanism for CTNNA1. Specifically, during the early stages of cell spreading, UBE2O-mediated ubiquitylation

of cytosolic CTNNA1 shifts its interaction preference toward vinculin, possibly by releasing its auto-inhibitory conformation, and promoting the establishment of cell-to-ECM adhesions.

The mechanosensitive interaction between CTNNA1 and vinculin at AJs has been well established in previous studies (Buckley et al, 2014; Ishiyama et al, 2013; Thomas et al, 2013; Yao et al, 2014; Yonemura et al, 2010). Within the E-cadherin/β-catenin

◀ **Figure 7. UBE2O selectively engages cytoplasmic CTNNA1 in FAs rather than AJs.**

(A) Immunoblot shows that pre-binding CTNNA1 with β-catenin reduces its interaction with UBE2O in vitro. (B) Immunoblot shows that co-expression with β-catenin decreases the UBE2O-mediated ubiquitylation of CTNNA1. (C) Immunoblot shows UBE2O-D3 predominantly ubiquitylates the cytosolic fraction of CTNNA1 in MDCK cells. MDCK cells were transfected with the indicated plasmids for 48 h. Cells were harvested for subcellular fractionation and western blot analysis according to the methods. (D) Representative micrographs show UBE2O selectively directs CTNNA1 to mediate FAs rather than AJs. The indicated MDCK cells co-stained for CTNNB1 and zyxin 8 h after plating on FN-coated 8-cell chamber slides; the left panel is a zoom-in of the box in the right image, showing the zyxin-containing FAs; white arrows point to locations of AJs. The graphs on the right are the quantifications of the number of FAs per cell (vinculin channel), average FA size (vinculin channel) of the indicated MEFs. Total number of cells analyzed: siNC (9), siUBE2O_1# (10), Res_Vector (V) (10), Res_UBE2O-WT (11), Res_UBE2O-CS (10). Error bars indicate mean ± SEM. Significance among multiple groups was determined using ANOVA followed by Tukey's post hoc test. All experiments were repeated at least twice, one representative result is shown. Source data are available online for this figure.

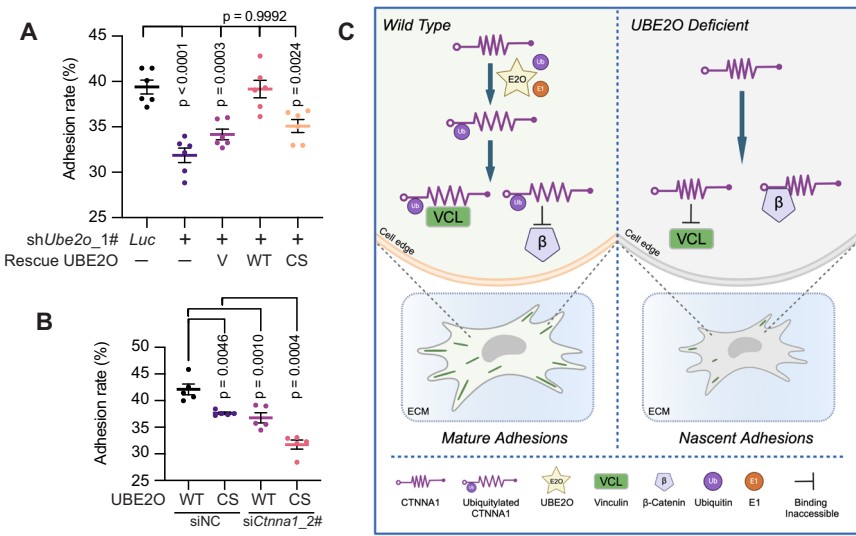

**Figure 8. UBE2O-CTNNA1 axis is required for cell adhesion to the ECM.**

(A) Adhesion assays show that the catalytic activity of UBE2O is responsible for MEFs' adhesion to the ECM during the early stages of cell spreading, $n = 6$ technical replicates in independent experiments. (B) Adhesion assays show UBE2O regulates cell-to-ECM adhesions via CTNNA1 in MEFs during the early stages of cell spreading, $n = 5$ technical replicates in independent experiments. Error bars indicate mean ± SEM. Significance among multiple groups was determined using ANOVA followed by Tukey's post hoc test. (C) Mechanism diagram of UBE2O-mediated CTNNA1 ubiquitylation in regulating cell-to-ECM adhesions. UBE2O selectively interacts with and ubiquitylates cytosolic CTNNA1. This ubiquitylation of CTNNA1 decreases its interaction with β-catenin while allowing its interaction with vinculin, thereby promoting the maturation of FAs and facilitating cell extension and cell adhesion to the ECM during the early stages of cell spreading. Consequently, this process selectively engages CTNNA1 in cell-to-ECM adhesions, rather than cell-to-cell adhesions. This diagram was created in BioRender. All experiments were repeated twice, one representative result is shown. Source data are available online for this figure.

complex at AJs, CTNNA1 binds vinculin in response to mechanical forces, forming a mechanoresponsive linkage between cadherins and the actin cytoskeleton. This interaction creates a self-reinforcing system that strengthens cell-to-cell adhesion. In our study, we revealed a distinct regulatory mechanism governing CTNNA1 function when it is not incorporated into AJ complexes. We found that UBE2O selectively interacts with and ubiquitylates cytosolic CTNNA1 (rather than the β-catenin-bound complex). This post-translational modification promotes the preferential interaction with vinculin over β-catenin, thereby leading to its recruitment to adhesion structures rather than AJs. Collectively, these findings delineate a novel ubiquitin-dependent pathway that complements the canonical mechanosensitive pathway at AJs.

In mammalian cells, phosphorylation of CTNNA1 predominantly localizes at cell-to-cell contacts and strengthens cell-to-cell adhesions, without affecting CTNNA1-vinculin interactions (Escobar et al, 2015). In contrast, our study demonstrates that UBE2O-catalyzed ubiquitylation facilitates CTNNA1-vinculin interaction,

thereby promoting cell-to-ECM adhesions maturation. The independence of these two regulatory mechanisms—ubiquitylation and phosphorylation—enables cells to efficiently select between adhesion pathways. Notably, the basal level of CTNNA1 ubiquitylation (Fig. 2D, lane 1) is relatively lower than its phosphorylation (Fig. 3B–D) under standard culturing conditions, suggesting that specific stimuli are required to activate the ubiquitylation of CTNNA1. This observation warrants further investigation to delineate the regulatory mechanisms involved. To address this, methodological advancements—such as the generation of higher-affinity CTNNA1 antibodies or implementation of ultrasensitive proteomics methods—will be critical for the identification and quantification of endogenous ubiquitylated CTNNA1. Such refinements may unveil context-dependent ubiquitylation events that are currently undetectable.

Although we identified several UBE2O-mediated ubiquitylation sites on CTNNA1 through mass spectrometry, CTNNA1 mutants with mutations in specific lysine residues showed no significant

resistance to UBE2O-mediated ubiquitylation. This could suggest compensatory ubiquitylation at alternate lysine residues or even other amino acids when primary sites are mutated, which aligns with the "alternative ubiquitylation theory" concept. This is further corroborated by our findings with the lysine-deficient, CTNNA1-KO mutant (Fig. 2F; Appendix Fig. S2E), highlighting the functional importance of ubiquitylation in regulating CTNNA1 dynamics.

Finally, the regulation of ubiquitylation is counterbalanced by deubiquitylases (DUBs), which remove ubiquitin molecules from substrate proteins to maintain substrate protein homeostasis (Lange et al, 2022). Our CTNNA1 interaction screen identified several DUBs, and previous work has shown that UBE2O can form complexes with specific DUB to collaboratively regulate target-mediated functions (Huang et al, 2022). Therefore, DUBs may play a critical role in regulating UBE2O-mediated CTNNA1 ubiquitylation and the overall maturation of FAs. Further investigations are needed to explore this regulatory interplay.

In conclusion, cell adhesions are dynamic structures regulated by intricate molecular machinery. Our findings elucidate a novel regulatory mechanism in which UBE2O selectively ubiquitylates cytosolic CTNNA1 to enhance its interaction with vinculin, rather than β-catenin, to promote FAs maturation, cell extension and ECM adhesion during the early stages of cell spreading. This mechanism preferentially directs CTNNA1 towards cell-to-ECM adhesions (Fig. 8C). However, key questions remain regarding the stimuli that activate UBE2O-mediated ubiquitylation of CTNNA1, the tissue-specific effects of this modification, and the potential regulatory roles of DUBs in this process. Addressing these questions in future studies will deepen our understanding of cell adhesion dynamics and their implications in health and disease.

## Methods

### Reagents and tools table

| Reagent/resource | Reference or source | Identifier or catalog number |
|---|---|---|
| **Experimental models** | | |
| HEK293T (*H. sapiens*) | ATCC | CRL-3216™ |
| Expi293F (*H. sapiens*) | Gibco | A14527 |
| U-2 OS (*H. sapiens*) | ATCC | HTB-96™ |
| MEF (*M. musculus*) | This study | N/A |
| MDCK (*Canis lupus familiaris*) | Cell bank of the Chinese Academy of Sciences | NBL-2 |
| BL21 (DE3) (*E. coli*) | Zhang et al, 2017 | N/A |
| **Recombinant DNA** | | |
| pLV-Flag-CTNNA1 | This study | N/A |
| pCR3.1-Myc-CTNNA1 | This study | N/A |
| pCR3.1-HA-CTNNA1 | This study | N/A |
| pGEX-6P-1-CTNNA1-Myc | This study | N/A |
| pGEX-6P-1-CTNNA1-V5 | This study | N/A |
| pCR3.1-Myc-CTNNA1 deletions | This study | N/A |

| Reagent/resource | Reference or source | Identifier or catalog number |
|---|---|---|
| pCR3.1-HA-CTNNA1-KR | This study | N/A |
| pCR3.1-HA-CTNNA1-5KR | This study | N/A |
| pCR3.1-Myc-CTNNA1-S641A | This study | N/A |
| pCR3.1-Myc-CTNNA1-S641D | This study | N/A |
| pCR3.1-HA-CTNNA1-KO | This study | N/A |
| pLV-Flag-CTNNA1-KO | This study | N/A |
| pLV-Flag-ResCTNNA1 | This study | N/A |
| pLV-Flag-ResCTNNA1-KO | This study | N/A |
| pCR3.1-Flag-CTNNA2 | This study | N/A |
| pCR3.1-UBE2O-Myc | Huang et al, 2022; Zhang et al, 2013b | N/A |
| pCR3.1-UBE2O-C1040S-mutated (CS)-Myc | Huang et al, 2022; Zhang et al, 2013b | N/A |
| pLV-Flag-UBE2O | Huang et al, 2022; Zhang et al, 2013b | N/A |
| pLV-Flag-UBE2O-CS | Huang et al, 2022; Zhang et al, 2013b | N/A |
| pLV-Flag-UBE2O deletions | Huang et al, 2022; Zhang et al, 2013b | N/A |
| pLV-Flag-UBE2O-TwinStrep | Huang et al, 2022; Zhang et al, 2013b | N/A |
| pCR3.1-6×His-Myc-Ubiquitin | Huang et al, 2022 | N/A |
| pET28a-His-HA-Ubiquitin | This study | N/A |
| pET28a-His-HA-Ubiquitin-KO | This study | N/A |
| pET28a-6×His-mUBA1 | This study | N/A |
| pLV-GFP-zyxin | This study | N/A |
| pCDNA3-Flag-CTNNB1 | This study | N/A |
| **Antibodies** | | |
| Goat anti-rabbit secondary HRP-conjugated antibody | Bethyl Laboratories | A120-101P |
| Goat anti-mouse secondary HRP-conjugated antibody | Bethyl Laboratories | A90-116P |
| Rabbit anti-Myc | Proteintech | 16286-1-AP |
| Rabbit anti-HA | Sigma-Aldrich | H6908 |
| Mouse anti-Flag tag | Sigma-Aldrich | F1804 |
| Rabbit anti-Flag | Merck Millipore | F7425 |
| Rabbit anti-UBE2O | Novus | NBP1-03336 |
| Rabbit anti-UBE2O | Cell Signaling Technology | 83393S |
| Rabbit anti-UBE2O | Bethyl Laboratories | A301-873A |
| Rabbit IgG | Bethyl Laboratories | P120-101 |
| Rabbit anti-CTNNA1 | Proteintech | 12831-1-AP |
| Mouse anti-CTNNA1 | Abcam | ab231306 |
| Mouse anti-streptag II | Merck Millipore | 71590-3cn |
| Rabbit anti-V5 tag | GeneTex | GTX117997 |
| Rabbit anti-GFP tag | Proteintech | 50430-2-AP |
| Mouse anti-β-Actin-HRP | Santa Cruz Biotechnology | SC-47778 |

| Reagent/resource | Reference or source | Identifier or catalog number |
|---|---|---|
| Rabbit anti-Vinculin | ThermoFisher Scientific | 700062 |
| Mouse anti-β-catenin (E-5) | Santa Cruz Biotechnology, | sc-7963 |
| Rabbit anti-β-catenin (D10A8) | Cell Signaling Technology | 8480 |
| Rabbit anti-Phospho-alpha 1 Catenin/CTNNA1 (Ser641) | Affinity Biosciences | AF8082 |
| Rabbit anti-GST tag | Invitrogen | PA1-982A |
| Mouse anti-α-Tubulin (DM1A) | Abcam | ab7291 |
| Rabbit anti-N-cadherin | Proteintech | 22018-1-AP |
| Rabbit anti-E-cadherin | Proteintech | 20874-1-AP |
| Mouse anti-GAPDH | Proteintech | 60004-1-Ig |
| Mouse anti-ubiquitin (P4D1) antibody | Santa Cruz Biotechnology | SC-8017 |
| Rabbit anti-zyxin | Affinity Biosciences | DF6858 |
| Alexa Fluor 488-conjugated goat anti-rabbit IgG (H + L) secondary antibody | Invitrogen | A11008 |
| Alexa Fluor 488-conjugated goat anti-mouse IgG (H + L) secondary antibody | Invitrogen | A11001 |
| Alexa Fluor 568-conjugated goat anti-rabbit IgG (H + L) secondary antibody | Invitrogen | A11011 |
| Alexa Fluor 568-conjugated goat anti-mouse IgG (H + L) secondary antibody | Invitrogen | A11004 |
| Alexa Fluor 647-conjugated goat anti-rabbit IgG (H + L) secondary antibody | Invitrogen | A32733 |
| **Oligonucleotides and other sequence-based reagents** | | |
| shRNAs | This study | Dataset EV2 |
| siRNAs | This study | Dataset EV2 |
| PCR primers | This study | Dataset EV2 |
| **Chemicals, enzymes and other reagents** | | |
| DMEM | Gibco | C11995500BT |
| FBS | Gibco | 10270106 |
| SMM 293-TII Expression Medium | Sino Biological | M293TII |
| PEI | Polysciences | 24765 |
| Phanta Max super-fidelity DNA polymerase | Vazyme | P505-d2 |
| Lipofectamine™ 3000 Transfection kit | Invitrogen | L3000075 |
| Lipofectamine™ RNAiMAX Transfection Reagent | Invitrogen | 13778150 |
| Puromycin dihydrochloride | MedChemExpress | HY-B1743A |
| Hygromycin B | MedChemExpress | HY-B0490-5 |
| Ampicillin sodium | Sigma-Aldrich | A9518 |
| Kanamycin sulfate | Mdbio Inc | K001 |
| Doxycycline hyclate | MedChemExpress | HY-N0565B |

| Reagent/resource | Reference or source | Identifier or catalog number |
|---|---|---|
| IPTG | MedChemExpress | HY-15921 |
| PMSF | Sangon Biotech | A610425 |
| DTT | Sigma-Aldrich | D9779 |
| Glycerol | Sigma-Aldrich | V900122 |
| Triton X-100 | Sigma-Aldrich | T9284 |
| 2-Mercaptoethanol | Sigma-Aldrich | M3148 |
| NEM | Sigma-Aldrich | E3876 |
| Imidazole | Sigma-Aldrich | V900153 |
| Biotin | Sigma-Aldrich | V900418 |
| Lambda PP | New England Biolabs | P0753S |
| Fibronectin | Yeasen | 40113ES03 |
| Acetonitrile (MS) | ThermoFisher Scientific | 51101 |
| 0.1% Formic Acid (v/v) in water (MS) | ThermoFisher Scientific | 85170 |
| Non-fat milk | Beyotime | P0216 |
| Beyoecl Star substrate | Beyotime | P0018AS |
| Protease Inhibitor Cocktail | Roche | 4693116001 |
| CTNNA1-Myc | This study | N/A |
| CTNNA1-V5 | This study | N/A |
| His-HA-Ubiquitin | This study | N/A |
| His-HA-Ubiquitin-KO | This study | N/A |
| His-mUBE1 | This study | N/A |
| Flag-UBE2O-WT-TwinStrep | This study | N/A |
| Flag-UBE2O-CS-TwinStrep | This study | N/A |
| His-GST-β-catenin protein | Sino Biological | 11279-H20B |
| 0.45 μm Filter | Taitan | TYLQ-0012 |
| 3 kDa MWCO Amicon Ultra-15 Centrifugal Filter | Merck Millipore | UFC9003 |
| 10 kDa MWCO Amicon Ultra-15 Centrifugal Filter | Merck Millipore | UFC901024 |
| 100 kDa MWCO Amicon Ultra-15 Centrifugal Filter | Merck Millipore | UFC910096 |
| Ni sepharose™ 6 Fast Flow | Cytiva | 17531803 |
| Superdex 75 increase 10/300 GL | Cytiva | 29148721 |
| Hiload® 16/600 Superdex® 200 pg | Cytiva | 28989335 |
| Strep-Tactin®XT 4Flow® resin | IBA Life Sciences | 2-5030-010 |
| Glutathione Resin | Genscript | L00206 |
| Anti-Flag M2 affinity gel | Sigma-Aldrich | A2220 |
| Pierce Protein A/G Magnetic Beads | ThermoFisher Scientific | 88803 |
| Ni-NTA agarose | QIAGEN | 30230 |
| Anti-c-Myc Magnetic Beads | ThermoFisher Scientific | 88843 |
| V5-Trap® magnetic agarose | Chromotek | v5tma |
| Pierce™ BCA Protein Assay Kit | ThermoFisher Scientific | 23225 |

| Reagent/resource | Reference or source | Identifier or catalog number |
|---|---|---|
| Subcellular Protein Fractionation Kit for Cultured Cells | ThermoFisher Scientific | 78840 |
| Nitrocellulose membrane | Cytiva | 10600003 |
| 8-cell chamber slides | Falcon | 354108 |
| 24-well Transwell plates | Corning | 3422 |
| DAPI | Cell Signaling Technology | 4083S |
| CoraLite®594-Phalloidin | Proteintech | PF00003 |
| Antifade Mounting Medium | Beyotime | P0126 |
| **Software** | | |
| MaxQuant v1.6.17.0 and v2.0.3.0) | Tyanova et al, 2016 | |
| Fiji v2.9.0 | Schindelin et al, 2012 | |
| GraphPad Prism v10.2.3 | https://www.graphpad.com | |
| DEP2 package | Feng et al, 2023; Zhang et al, 2018 | |
| **Other** | | |

## Plasmids, cell culture and transfection

The full-length mouse *Ctnna1* was amplified from cDNA of IB10 cells and subsequently cloned into CMV promoter-based Flag-, Myc- or HA-tagged vectors as described before (Zhang et al, 2008; Zhang et al, 2013b). For prokaryotic expression plasmids, the full-length mouse *Ctnna1* was cloned into C-terminal Myc- or V5-tagged pGEX-6P-1 vectors. The full-length human *CTNNA2* was amplified from cDNA of Expi293F cells and subsequently cloned into a CMV promoter-based Flag-tagged vector. The full-length and truncated versions of pCR3.1-UBE2O-Myc, pCR3.1-UBE2O-C1040S-mutated (CS)-Myc, pLV-Flag-UBE2O and pLV-Flag-UBE2O-CS plasmids have been described previously (Huang et al, 2022; Zhang et al, 2013b). To express and purify UBE2O protein, the full-length WT or CS human *UBE2O* was cloned into N-terminal Flag-tagged and C-terminal TwinStrep-tagged pLV vectors. For the truncated versions of CTNNA1, the indicated regions were cloned into N-terminal Myc-tagged pCR3.1 vector. The single KR and 5KR mutated CTNNA1 plasmids were mutated based on HA-mCTNNA1 plasmid and S641A, S641D mutated CTNNA1 plasmids were mutated based on Myc-mCTNNA1 plasmid using Phanta Max Super-Fidelity DNA Polymerase (Vazyme, P505-d2). The *Ctnna1-KO* (all lysine residues are replaced with arginine residues) fragment was synthesized from IGE Bio, China, and subsequently cloned into an N-terminal HA-tagged pCR3.1 vector or a Flag-tagged pLV vector. The siRNA-resistant CTNNA1-WT and -KO plasmids were mutated from pLV-Flag-mCTNNA1-WT or -KO plasmids. The pCR3.1-6×His-Myc-Ubiquitin-WT has been described previously (Huang et al, 2022). Mouse *Uba1* was amplified from cDNA of NIH3T3 cells and cloned into an N-terminal 6×His-tagged pET28a vector. The full-length human *ZYX* was amplified from ORF of CCSB-Broad Lentiviral Expression Library (AccessionList: BC008743) and cloned into N-terminal GFP-tagged pLV vector. Human *CTNNB1*

amplified from cDNA of HEK293T cells was cloned into Flag-tagged pcDNA3 vector. The lentiviral plasmids expressing specific shRNAs for human *UBE2O* have been described previously (Zhang et al, 2013b). Three mouse *Ube2o* shRNAs sequences from Sigma-Aldrich (MISSION® shRNA) were tested, and the effective shRNAs (sh*Ube2o* UTR-1: CCAGAGGTTTACAAGTTTCTA, sh*Ube2o* CDS-1: CGTCTGTTGAAGAAGCAAGTT) together with a shRNA sequence targeting *Luciferase* (CATTCTATCCTCTAGAGGATG) were cloned into FH1(NSR)UTP vector (Shi et al, 2021) to construct inducible shRNA plasmids. The siRNA kit genOFF hUBE2O_2500A for knockdown *UBE2O* was purchased from RiboBio, China, which has been described previously (Huang et al, 2022). The siRNA kits genOFF mCTNNA1 and cfaUBE2O to knockdown mouse *Ctnna1* and dog *UBE2O* were purchased from RiboBio, China. All plasmids were confirmed by DNA sequencing. The sequence of oligos used for plasmids construction and siRNAs are shown in Dataset EV2.

HEK293T (CRL-3216™) and U-2 OS (HTB-96™) cells were purchased from ATCC. Expi293F™ cells were obtained from Gibco (A14527). MDCK (NBL-2) cells were purchased from the cell bank of the Chinese Academy of Sciences. Mouse embryonic fibroblast cells (MEFs) were isolated from mouse embryos at embryonic day 13.5 (E13.5)-E14.5. MEF cells were immortalized by lentivirus-mediated expression of SV40. HEK293T, U-2 OS, MDCK (NBL-2) and MEF cells were routinely maintained in Dulbecco's Modified Eagle's Medium (DMEM, Gibco, C11995500BT) supplemented with 10% Fetal bovine serum (FBS, Gibco, 10270106). All cells except Expi293F were maintained as monolayers in a humidified atmosphere containing 5% $CO_2$ at 37 °C. Expi293F cells were maintained in SMM 293-TII Expression Medium (Sino Biological, M293TII). All cells were tested for mycoplasma once per month routinely using PCR-based method (Siegl et al, 2023).

Poly-ethylenimine (PEI, Polysciences, 24765-1 g) reagent, Lipofectamine™ 3000 Transfection kit (Invitrogen, L3000075; Fig. 7C) were used for plasmids transfection, and Lipofectamine™ RNAiMAX Transfection Reagent (Invitrogen, 13778150) was used for siRNA transfection. The media containing lentivirus particles were obtained from the culture supernatant of HEK293T cells. The HEK293T cells were co-transfected with target gene plasmids and three helper plasmids, pCMV-VSVG, pMDLg-RRE (gag/pol), and pRSV-REV. Lentiviruses were used to infect target cells, and stable cell lines were obtained after Puromycin dihydrochloride (Med-ChemExpress, HY-B1743A) or Hygromycin B (MedChemExpress, HY-B0490-5) selection as described before (Zhang et al, 2013b). For doxycycline-induced knockdown experiments, cells were treated with 500 ng/mL Doxycycline hyclate (MedChemExpress, HY-N0565B) for the indicated time periods.

## Western blot and antibodies

Cells were lysed in lysis buffer (0.5% NP40, 150 mM NaCl, 50 mM Tris pH 8.0, 1 mM EDTA, 10% Glycerol (Sigma-Aldrich, V900122)) supplemented with 1 × cOmplete™ Protease Inhibitor Cocktail (Roche, 4693116001) on ice for 30 min. The lysates were then centrifuged at 20,000 × g for 10 min at 4 °C. The supernatants were transferred into a new tube and the protein concentrations of the lysates were measured using the Pierce™ BCA Protein Assay Kit (ThermoFisher Scientific, 23225). Equivalent amounts of proteins were boiled with 1 × LDS loading buffer for 5 min at

95 °C, resolved by SDS-PAGE and transferred to nitrocellulose membrane (Cytiva, 10600003) for immunoblotting. Membranes were blocked with 5% non-fat milk (Beyotime, P0216) in TBST (50 mM Tris-HCl pH 7.6, 150 mM NaCl, 0.1% Tween 80) and incubated with the indicated primary antibodies overnight at 4 °C. The membranes were washed three times with TBST by the addition of goat anti-rabbit (Bethyl Laboratories, A120-101P; 1:10,000), or goat anti-mouse (Bethyl Laboratories, A90-116P; 1:10,000) secondary HRP-conjugated antibody for 1 h at room temperature. After three times washes, images were taken using ChemiDoc Imaging systems from Bio-Rad using BeyoECL Star substrate (Beyotime, P0018AS).

Primary antibodies used in western blot included rabbit anti-Myc (Proteintech, 16286-1-AP; 1:5000), rabbit anti-HA (Sigma-Aldrich, H6908; 1:5000), mouse anti-Flag tag (Sigma-Aldrich, F1804; 1:5000), rabbit anti-Flag (Merck millipore, F7425; 1:1500; Appendix Fig. S1A), rabbit anti-UBE2O (Novus, NBP1-03336; 1:2000; Fig. EV2A), rabbit anti-UBE2O (Cell signaling technology, 83393S; 1:2000), rabbit anti-CTNNA1 (Proteintech, 12831-1-AP; 1:2000), mouse anti-Strep•Tag® II (Merck millipore, 71590-3cn; 1:4000), rabbit anti-V5 tag (GeneTex, GTX117997; 1:5000), rabbit anti-GFP tag (Proteintech, 50430-2-AP; 1:7000), mouse anti-β-Actin-HRP (Santa Cruz Biotechnology, SC-47778; 1:5000), rabbit anti-Vinculin (ThermoFisher Scientific, 700062; 1:1000), mouse anti-β-catenin (E-5) (Santa Cruz Biotechnology, sc-7963; 1:1000), rabbit anti-β-catenin (D10A8) (Cell signaling technology, 8480; 1:1,000; Fig. EV4E), rabbit anti-Phospho-alpha 1 Catenin/CTNNA1 (Ser641) (Affinity Biosciences, AF8082; 1:1000), rabbit anti-GST tag (Invitrogen, PA1-982A; 1:3000), mouse anti-α-Tubulin (DM1A) (Abcam, ab7291; 1:4000), rabbit anti-N-cadherin (Proteintech, 22018-1-AP; 1:2000), rabbit anti-E-cadherin (Proteintech, 20874-1-AP; 1:2000) and mouse anti-GAPDH (Proteintech, 60004-1-Ig; 1:10,000).

## Protein expression and purification

For recombinant protein expression in bacteria, pGEX-6P-1-mCTNNA1-Myc, pGEX-6P-1-mCTNNA1-V5, pET28a-His-HA-Ubiquitin and pET28a-His-HA-Ubiquitin-KO plasmids were transformed into *E. coli* BL21 (DE3) cells (Zhang et al, 2017). The transformed cells were cultured in LB medium containing 100 μg/mL Ampicillin sodium (Sigma-Aldrich, A9518) or 50 μg/mL Kanamycin sulfate (Mdbio Inc, K001-10g) at 37 °C to an $OD_{600}$ of 0.6–0.8, and then induced with 0.5 mM Isopropyl β-D-thiogalactoside (IPTG; MedChemExpress, HY-15921) at 16 °C for 18 h. The bacteria pellets were resuspended in bacteria lysis buffer (50 mM Tris-HCl pH 7.4, 200 mM NaCl, 1 mM Phenylmethyl sulfonyl fluoride (PMSF; Sangon Biotech, A610425), 2 mM DL-Dithiothreitol (DTT; Sigma-Aldrich, D9779) and 5% glycerol) and lysed by sonication, then clarified by centrifugation at $30,000 \times g$ at 4 °C for 1 h. The supernatants were filtered through a 0.45-μm filter (Taitan, TYLQ-0012). Cleared lysates of CTNNA1 were loaded onto Glutathione Resin (Genscript, L00206). The bound proteins were washed with 100 mL of washing buffer (50 mM Tris-HCl pH 7.4, 200 mM NaCl, 2 mM DTT, 5% glycerol). Then, the GST tag was removed by incubating with PreScission protease (homemade) overnight on a rotation wheel at 4 °C. The cleaved fractions were collected and concentrated using a 10 kDa molecular weight cut-off (MWCO) Amicon Ultra-15 Centrifugal Filter (Merck Millipore,

UFC901024). Cleared lysates of WT or KO ubiquitin were loaded onto Ni Sepharose™ 6 Fast Flow (Cytiva, 17531803). The column was washed with 150 mL $His_6$ washing buffer (50 mM Tris-HCl pH 7.4, 200 mM NaCl, 20 mM imidazole (Sigma-Aldrich, V900153), 1 mM DTT). Then, the ubiquitin protein was eluted with $His_6$ elution buffer (50 mM Tris-HCl pH 8.0, 250 mM imidazole, 200 mM NaCl, 1 mM DTT), followed by concentration in a 3 kDa MWCO Amicon Ultra-15 Centrifugal Filter (Merck Millipore, UFC9003). The concentrated proteins were loaded onto a Superdex 75 increase 10/300 GL (Cytiva, 29148721) and gel filtrations were carried out in the buffer containing 50 mM Tris-HCl pH 7.4, 200 mM NaCl, 2 mM DTT. The His-mUBE1 protein was purified as previously reported (Liu et al, 2021). The peak fractions were pooled and concentrated, then the protein in 50% glycerol was snap-frozen in liquid nitrogen and stored at −80 °C.

To express and purify UBE2O protein, pLV-Flag-UBE2O-WT-TwinStrep or pLV-Flag-UBE2O-CS-TwinStrep plasmids were transfected into suspended Expi293F cells using PEI for 48 h. Cell pellets washed with PBS were lysed in lysis buffer supplemented with 1× cOmplete™ Protease Inhibitor Cocktail and 1 mM PMSF at 4 °C for 30 min, followed by clearance by centrifugation at $20,000 \times g$ at 4 °C for 2 h. The supernatant was filtered through a 0.45-μm filter and loaded onto Strep-Tactin®XT 4Flow® resin (IBA Life Sciences, 2-5030-010). The bound protein was washed with washing buffer, then eluted with Strep elution buffer (50 mM Tris-HCl pH 7.4, 50 mM biotin (Sigma-Aldrich, V900418), 0.5 mM DTT, 10% glycerol), followed by concentration in a 100 kDa MWCO Amicon Ultra-15 Centrifugal Filter (Merck Millipore, UFC910096). The concentrated proteins were loaded onto a HiLoad® 16/600 Superdex® 200 pg (Cytiva, 28989335) and gel filtrations were carried out in the buffer containing 50 mM Tris-HCl pH 7.4, 200 mM NaCl, 1 mM DTT. The peak fractions were pooled and concentrated, then the protein in 10% glycerol was snap-frozen in liquid nitrogen and stored at −80 °C.

## In vivo ubiquitylation assay by immunoprecipitation

In vivo ubiquitylation assay by anti-Flag immunoprecipitation (Flag-IP) was conducted as previously described (Huang et al, 2022). Briefly, HEK293T cells transfected with indicated plasmids for 36 h were washed twice with ice-cold PBS containing 10 mM N-Ethylmaleimide (NEM, Sigma-Aldrich, E3876). The cells were lysed in radioimmunoassay buffer (20 mM $NaH_2PO_4$-$Na_2HPO_4$ pH 7.4, 150 mM NaCl, 1% Triton X-100, and 0.5% sodium deoxycholate) supplemented with 1% SDS, 1× cOmplete™ Protease Inhibitor Cocktail and 10 mM NEM at 4 °C for 30 min. The lysates were sonicated by Bioruptor Pico with ten cycles of 30 s on and off, and then heated at 95 °C for 5 min. Subsequently, the cell lysates were diluted to 0.1% SDS with radioimmunoassay buffer and cleared by centrifugation at $20,000 \times g$ at 4 °C for 10 min. The supernatants were incubated with pre-washed anti-Flag M2 affinity gel (Sigma-Aldrich, A2220) for 90 min at 4 °C, followed by three times washes with radioimmunoassay buffer. The bound proteins were eluted with 2× LDS loading buffer at 42 °C for 15 min and separated with SDS-PAGE and analyzed by immunoblotting.

For in vivo ubiquitylation assay by ubiquitin antibody (P4D1), a similar method to that of Flag-IP-based ubiquitylation assay was performed. However, in this case, the supernatants were incubated with mouse anti-ubiquitin (P4D1) antibody (Santa Cruz Biotechnology,

SC-8017) overnight at 4 °C on a rotation wheel. Subsequently, they were incubated with pre-washed Pierce™ Protein A/G Magnetic Beads (ThermoFisher Scientific, 88803) for 90 min at 4 °C followed by three times washes with radioimmunoassay buffer. The bound proteins were eluted with 2 × LDS loading buffer at 42 °C for 15 min and separated with SDS-PAGE and analyzed by immunoblotting.

## In vivo ubiquitylation assay by nickel pull-down

In vivo ubiquitylation assay by nickel pull-down (Ni-PD) was performed as previously described (Zhang et al, 2013a). In brief, HEK293T cells transfected with the indicated plasmids were washed twice with ice-cold PBS containing 10 mM NEM. The cells were lysed in a denaturing buffer (8 M urea, 0.1 M $NaH_2PO_4$-$Na_2HPO_4$, 10 mM Tris-HCl pH 7.0, 10 mM imidazole) supplemented with 10 mM 2-Mercaptoethanol (Sigma-Aldrich, M3148) at room temperature for 15 min. The lysates were centrifuged at $20,000 \times g$ for 10 min at room temperature and the supernatants were incubated with pre-washed Ni-NTA agarose (QIAGEN, 30230) at room temperature for 2 h, followed by three times washes with denaturing buffer. The bound proteins were eluted with 2 × LDS loading buffer for 15 min at 42 °C before being separated with SDS-PAGE and analyzed by immunoblotting.

## In vitro interaction and ubiquitylation assay

For in vitro CTNNA1 and UBE2O interaction assay, 0.1 μM purified CTNNA1-V5 was mixed with 0.1 μM purified Flag-UBE2O-TS protein in lysis buffer and then incubated with Strep-Tactin®XT 4Flow® resin at 4 °C for 1.5 h. Alternatively, 0.1 μM purified Flag-UBE2O-TS was mixed with 0.1 μM purified CTNNA1-Myc in lysis buffer then incubated with Pierce™ Anti-c-Myc Magnetic Beads (ThermoFisher Scientific, 88843) at 4 °C for 1.5 h. After washing three times with the lysis buffer, proteins bound to beads were eluted by boiling with 2 × LDS loading buffer at 95 °C for 15 min and then subjected to immunoblotting assay.

For in vitro CTNNA1-β-catenin complex and UBE2O interaction assay, 0.1 μM CTNNA1-V5 protein was incubated with or without 25 nM or 50 nM His-GST-β-catenin protein (Sino Biological, 11279-H20B) in lysis buffer on a rotation wheel at 4 °C overnight. V5-Trap® magnetic agarose (Chromotek, v5tma) beads were pre-blocked with 2% bovine serum albumin (BSA) in lysis buffer at room temperature for 1 h and then used to enrich CTNNA1 or CTNNA1-β-catenin complex. After washing three times with the lysis buffer, 0.1 μM Flag-UBE2O-TS protein was incubated with beads in lysis buffer at 4 °C for 1.5 h. After washing three times with the lysis buffer, proteins bound to beads were eluted by boiling with 2 × LDS loading buffer at 95 °C for 15 min and then subjected to immunoblotting assay.

For in vitro ubiquitylation assay, 2 μM purified mCTNNA1 was incubated at 37 °C for 1 h with 50 μM His-HA-ubiquitin, 100 nM mE1 and 2 μM UBE2O in ubiquitylation reaction buffer containing 2 mM ATP, 5 mM $MgCl_2$, 50 mM NaCl, and 50 mM Tris-HCl pH 8.0. The reaction was quenched with 1 × LDS loading buffer for 5 min at 95 °C.

## In vitro dephosphorylation assay

HEK293T cells transfected with Myc-CTNNA1 plasmid for 36 h were harvested and lysed with lysis buffer supplemented with

$1 \times cOmplete^{TM}$ Protease Inhibitor Cocktail. The lysates were centrifuged at $20,000 \times g$ for 10 min at 4 °C. The supernatants were incubated with Pierce™ Anti-c-Myc Magnetic Beads for 90 min at 4 °C followed by three times washes with lysis buffer. The beads were then incubated with or without Lambda Protein Phosphatase (Lambda PP, New England Biolabs, P0753S) according to the manufacturer's protocol at 30 °C for 30 min in a table shaker, followed by 3 times washes with lysis buffer to remove the phosphatase. Subsequently, in vitro ubiquitylation reactions were performed by incubating beads at 37 °C for 1 h with 50 μM His-HA-ubiquitin, 100 nM mE1 and 2 μM UBE2O in ubiquitylation reaction buffer with or without ATP in a table shaker. Proteins bound to beads were eluted and the reactions were quenched with 1 × LDS loading buffer for 15 min at 95 °C.

## Immunoprecipitation and proteomics sample preparation

Cells transfected with the indicated plasmids for 36 h were washed twice with ice-cold PBS and lysed by lysis buffer supplemented with $1 \times cOmplete^{TM}$ Protease Inhibitor Cocktail. The lysates were centrifuged at $20,000 \times g$ for 10 min at 4 °C. The supernatants were incubated with pre-washed anti-Flag M2 affinity gel for 90 min at 4 °C followed by three times washes with lysis buffer. For endogenous co-immunoprecipitation, the protein concentration of the lysates were measured by the Pierce™ BCA Protein Assay Kit. The supernatants of MEFs or HaCaT cell lysates were incubated with rabbit anti-UBE2O (Novus, NBP1-03336, Fig. 1B), rabbit anti-UBE2O (Bethyl Laboratories, A301-873A, Fig. 1C), or purified rabbit IgG (Bethyl Laboratories, P120-101) antibody overnight at 4 °C on a rotation wheel. The supernatants of shLuciferase/shUbe2o MEF cell lysates were incubated with rabbit anti-CTNNA1 (Proteintech, 12831-1-AP, Fig. EV3E), or purified rabbit IgG antibody overnight at 4 °C on a rotation wheel. Subsequently, the reaction was incubated with pre-washed Pierce™ Protein A/G Magnetic Beads for 90 min at 4 °C, followed by three times washes with lysis buffer. For co-immunoprecipitation analysis, beads were boiled with 2 × LDS loading buffer for 15 min at 95 °C for subsequent analysis by immunoblotting.

To identify CTNNA1 interactors in HEK293T cells or Ube2o-deficient MEFs by mass spectrometry-based proteomic method, three technical repeats were performed and on-bead digestion with trypsin was performed as described before (Zhang et al, 2017).

For ubiquitylated CTNNA1 interactome identification, Flag-UBE2O-D3 was transfected into HEK239T cells for 36 h. The cells were washed twice with ice-cold PBS and lysed by lysis buffer supplemented with $1 \times cOmplete^{TM}$ Protease Inhibitor Cocktail at 4 °C for 30 min. The lysate was centrifuged at $20,000 \times g$ for 10 min at 4 °C. Subsequently, the supernatant was incubated with anti-Flag M2 affinity gel for 90 min at 4 °C. After three times washes with lysis buffer, beads were distributed equally to 3 tubes to catalyze the control, CTNNA1 and ubiquitylated CTNNA1 (CTNNA1_Ub) baits. The CTNNA1_Ub bait was catalyzed by adding 50 μM His-HA-ubiquitin, 100 nM mE1, 2 μM purified CTNNA1-V5 in ubiquitylation reaction buffer to beads and incubating at 37 °C for 1 h. The same reaction without CTNNA1-V5 (ctrl) or ATP (CTNNA1) in the ubiquitylation reaction was conducted to prepare the control or CTNNA1 bait. The reaction mixtures were centrifuged at $4000 \times g$ for 2 min at 4 °C, and the supernatants were collected to incubate with V5-Trap® magnetic agarose at 4 °C for 2 h on a rotation wheel,

followed by 3 times washes with lysis buffer. The indicated cell lysate was incubated with the beads overnight at 4 °C on a rotation wheel, followed by four times washes with lysis buffer. For western blot identification, beads were boiled with 2 × LDS loading buffer for 15 min at 95 °C for subsequent analysis by immunoblotting. For proteomics sample preparation, three repeats were performed and on-bead digestion with trypsin was performed as described before (Zhang et al, 2017).

For UBE2O full-length mediated CTNNA1 ubiquitylation sites identification, Expi293F cells transfected with Flag-CTNNA1, 6 × His-Myc-Ubiquitin and UBE2O-CS-Myc plasmids (control group), or Flag-CTNNA1, 6 × His-Myc-Ubiquitin and UBE2O-WT-Myc plasmids (ubi group) were harvested to perform the anti-Flag immunoprecipitation followed by nickel pull-down according to the previous report (Huang et al, 2022). For proteomics analysis, in-gel digestion with trypsin was performed. For UBE2O-D3 mediated CTNNA1 ubiquitylation sites identification, HEK293T cells transfected with Flag-UBE2O-D3 and Myc-CTNNA1 were harvested to perform the anti-Flag immunoprecipitation followed by in-gel digestion with trypsin. 2-chloroacetamide instead of iodoacetamide was used for alkylation for ubiquitylation sites identification.

## Mass spectrometry measurement and data analysis

For CTNNA1 ubiquitylation sites identification, tryptic peptides were separated using 60 min of total data collection time (2 min of 4% to 8%, 43 min of 8% to 28%, 8 min of 28% to 36%, 2 min of 36% to 100% and 5 min of 100% of buffer B (80% acetonitrile (ThermoFisher Scientific, 51101)) in 0.1% formic acid (v/v) in water (ThermoFisher Scientific, 85170) with a 300 nL/min flow using an Easy-nLC 1200 connected online to a Fusion Lumos mass spectrometer (Thermo-Fisher Scientific). Scans were collected in data-dependent top-speed mode with dynamic exclusion at 60 s. For CTNNA1 and ubiquitylated CTNNA1 interactors identification, tryptic peptides were separated using 140 min of total data collection (100 min of 2% to 22%, 20 min of 22% to 28%, and 12 min of 28% to 36%, 2 min of 36% to 100% and 6 min of 100% buffer B with a 300 nL/min flow using an Easy-nLC 1200 connected online to a Fusion Lumos mass spectrometer. Scans were collected in data-dependent top-speed mode with dynamic exclusion at 90 s. Raw data were analyzed using MaxQuant (version 1.6.17.0, version 2.0.3.0 and version 2.4.2.0) searched against the human FASTA database or the mouse FASTA database according to the cell species, with label-free quantification and match between runs functions enabled (Tyanova et al, 2016). For ubiquitylation sites identification, mCTNNA1 FASTA (P26231) was used and the (GlyGly) remnants search was enabled. The output protein lists were analyzed and visualized using DEP2 package (Feng et al, 2023; Zhang et al, 2018). The accession number for the raw mass spectrometry data reported in this paper is ProteomeXchange: PXD058956.

## Subcellular fractionation

HEK293T and MDCK cells were transfected with Flag-CTNNA1, Myc-UBE2O-D3 and 6 × His-Myc-Ubiquitin plasmids for 48 h. Cells were harvested and distributed equally for the preparation of subcellular and total extracts. Cytoplasmic and membrane extracts were isolated using the Subcellular Protein Fractionation Kit for Cultured Cells (ThermoFisher Scientific, 78840). For the total

extract, cells were lysed in lysis buffer supplemented with 1% SDS and 1 × cOmplete™ Protease Inhibitor Cocktail at 4 °C for 30 min. The lysates were sonicated by Biorupter Pico with ten cycles of 30 s on and off, and then heated at 95 °C for 5 min.

## Immunofluorescence microscopy

For immunofluorescence, cells were plated on fibronectin (FN; Yeasen, 40113ES03)-coated eight-cell chamber slides (Falcon, 354108) for 6 h (MEFs) or 8 h (MDCK) and fixed with 4% PFA. Blocking and permeabilization were performed using 10% FBS and 0.3% Triton X-100 (Sigma-Aldrich, T9284) in PBS for 1 h at RT. Immunostaining was performed with primary antibodies against rabbit anti-vinculin (ThermoFisher Scientific, 700062; 1:200), rabbit anti-zyxin (Affinity Biosciences, DF6858; 1:100), rabbit anti-UBE2O (Novus, NBP1-03336; 1:100), mouse anti-CTNNA1 (Abcam, ab231306; 1:200), mouse anti-β-catenin (E-5) (Santa Cruz Biotechnology, sc-7963; 1:200) and rabbit anti-E-cadherin (Pro-teintech, 20874-1-AP; 1:200) overnight at 4 °C. The slides were washed three times with PBS followed by the addition of Alexa Fluor 488-conjugated goat anti-rabbit IgG (H + L) secondary antibody (Invitrogen, A11008; 1:500), Alexa Fluor 488-conjugated goat anti-mouse IgG (H + L) secondary antibody (Invitrogen, A11001; 1:500; Fig. 5F), Alexa Fluor 568-conjugated goat anti-mouse IgG (H + L) secondary antibody (Invitrogen, A11004; 1:500), Alexa Fluor 568-conjugated goat anti-rabbit IgG (H + L) secondary antibody (Invitrogen, A11011; 1:500; Fig. 5F), or Alexa Fluor 647-conjugated goat anti-rabbit IgG (H + L) secondary antibody (Invitrogen, A32733; 1:500) for 1 h at RT in the dark. After washing the secondary antibodies three times with PBS, CoraLite®594-Phalloidin (Proteintech, PF00003) was added for 20 min at RT, or DAPI (Cell signaling technology, 4083S; Fig. 7D; Appendix Fig. S3C) was added for 5 min. After washing four times with PBS, Antifade Mounting Medium (Beyotime, P0126) was added to mount the coverslips. Images of Figs. 4A and 5F were taken at room temperature with an Elyra 7 Lattice SIM using a 40×/1.4 oil objective controlled by Zen imaging software (Zeiss). Other images were taken at room temperature with a Zeiss LSM800 confocal microscope using a 40×/1.3-NA oil objective controlled by Zen imaging software (Zeiss).

## Image analyses and quantifications

A built-in Fiji (Schindelin et al, 2012) macro, version 2.9.0, was used to analyze the confocal and western blot images. To measure the number of FAs per cell and average FA size, the confocal images were subjected to an intensity threshold to select the FAs using the vinculin or zyxin channel (as indicated in the figure legends), and the "Analyze Particles" tool was used. Because the cell sizes indicated by zyxin, vinculin and phalloidin channels are nearly identical in MEFs and MDCK cells (Figs. EV1E,F and EV4D), zyxin or vinculin channel was alternatively used to measure cell size when the measurement using phalloidin channel was interfered by the background nonspecific staining due to the differences of the slides. To measure the area of the cell, the "Measure" tool was used after Gaussian blurring, thresholding and filling holes of the vinculin, zyxin or phalloidin channel (as indicated in the figure legends). For measurements in Figs. 4A and 5F, the "Plot Profile" tool was used. Analyses of the integrated density of the blots in Fig. 7A,B and

Appendix Fig. S3A were performed using the "Measure" tool after subtracting background and inverting.

## Transwell assay

In all, $1 \times 10^5$ cells were suspended in 200 μL of DMEM and seeded into the upper chambers of 24-well Transwell plates (Corning, 3422). Then, 600 μL of DMEM supplemented with 10% FBS was added to the lower chambers, and the cells were incubated at 37 °C for 12 h. Following incubation, the lower chambers were fixed, and the cells were stained with crystal violet. Images were captured using a microscope, and the numbers of migrated cells in the lower chamber were calculated using Fiji.

## Adhesion assay

For cell adhesion assays, 96-well plates were coated with fibronectin at 4 °C overnight. The plates were then blocked with 1% inactive BSA at 37 °C for 1 h. After washing the plates twice with DMEM, $2 \times 10^4$ cells were seeded into the 96-well plates and incubated at 37 °C for 30 min. Cells were either washed three times with PBS prior to detection or left unwashed. The CCK-8 solution (Beyotime, C0043) was used to detect live attached cells. The adhesion rate was calculated by dividing the absorbance at 450 nm of the washed group by the absorbance of the non-washed group.

## Statistical analysis

The statistical analysis was carried out using GraphPad Prism (version 10.2.3). All quantifications represent the mean ± SEM. Comparisons were performed by ANOVA followed by Tukey's post hoc test. Differences were considered to be statistically significant from a *P* value < 0.05.

# Data availability

The mass spectrometry data underlying Figs. 1A, 5B,C, EV3B,E and Appendix Fig. S2B, C have been deposited to the ProteomeXchange Consortium (https://proteomecentral.proteomexchange.org) via the iProX partner repository (Chen et al, 2022) with the dataset identifier: PXD058956.

The source data of this paper are collected in the following database record: biostudies:S-SCDT-10_1038-S44319-025-00585-4.

# Peer review information

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

## Acknowledgements

We would like to extend our gratitude to all members of the Zhang lab for their valuable discussions. This study was supported by grants from the National Natural Science Foundation (32370752), Guangdong Basic and Applied Basic Research Project (2024A1515012344), and the Science and Technology Planning Project of Guangdong Province (2023B1212060050, 2023B1212120009). *Declaration of generative AI and AI-assisted technologies in the writing process*: During the preparation of this work, the authors used

GPT-4o mini (https://openai.com) via Sider in order to fix grammar errors and improve the clarity, coherence and overall readability of the text. After using this tool, the authors reviewed and edited the content as needed and take full responsibility for the content of the published article.

## Author contributions

**Dan Xiang**: Conceptualization; Resources; Data curation; Formal analysis; Validation; Investigation; Visualization; Methodology; Writing—original draft; Project administration; Writing—review and editing. **Wenfeng Wu**: Resources; Investigation. **Ruona Shi**: Resources; Investigation. **Xiaoxiao Tang**: Investigation. **Xiaofei Zhang**: Conceptualization; Resources; Supervision; Funding acquisition; Project administration; Writing—review and editing.

Source data underlying figure panels in this paper may have individual authorship assigned. Where available, figure panel/source data authorship is listed in the following database record: biostudies:S-SCDT-10_1038-S44319-025-00585-4.

## Disclosure and competing interests statement

The authors declare no competing interests.

# Expanded View Figures

**Figure EV1. Catalytic activity of UBE2O is required for adhesion maturation.**

(**A**) CTNNA1 is required for FAs maturation. Representative micrographs of MEFs transfected with siRNAs targeting *Ctnna1* (si*Ctnna1*) or a negative control (siNC) for 48 h. The cells were stained for zyxin after spreading on fibronectin (FN)-coated 8-cell chamber slides for 6 h. (**B**) Western blot analysis of MEFs stably expressing GFP-vector or GFP-zyxin. (**C, D**) UBE2O is required for FAs maturation in MEFs. (**C**) Representative micrographs of MEFs treated with doxycycline for 48 h to express shRNAs targeting *Ube2o* UTR region (1#) or CDS region (2#), alongside a negative control targeting luciferase (sh*Luc*). After 6 h of spreading on FN-coated 8-cell chamber slides, cells were stained for zyxin. (**D**) Quantifications of the number of FAs per cell (zyxin channel), average FA size (zyxin channel) and cell area (zyxin channel) of the *Ube2o* knockdown MEFs 6 h after plating on FN-coated 8-cell chamber slides ($n = 7$ cells in all cases). Error bars indicate mean ± SEM. Significance among multiple groups was determined using ANOVA followed by Tukey's post hoc test. (**E**) The catalytic activity of UBE2O is required for FAs maturation and cell extension in MEFs. Representative micrographs of *Ube2o* knockdown MEFs with or without rescuing the expression of UBE2O-WT, UBE2O-CS mutant, or a control vector. All slides were stained for zyxin to visualize FAs and phalloidin to visualize cell area after 6 h of spreading on FN-coated 8-cell chamber slides. (**F**) UBE2O is responsible for FAs maturation in MDCK cells and is catalytic activity-dependent. Representative micrographs of *UBE2O* knockdown MDCK cells with or without rescuing the expression of UBE2O-WT, UBE2O-CS mutant or a control vector. All slides were stained for zyxin and phalloidin after 8 h of spreading. Experiments were repeated at least twice, one representative result is shown.

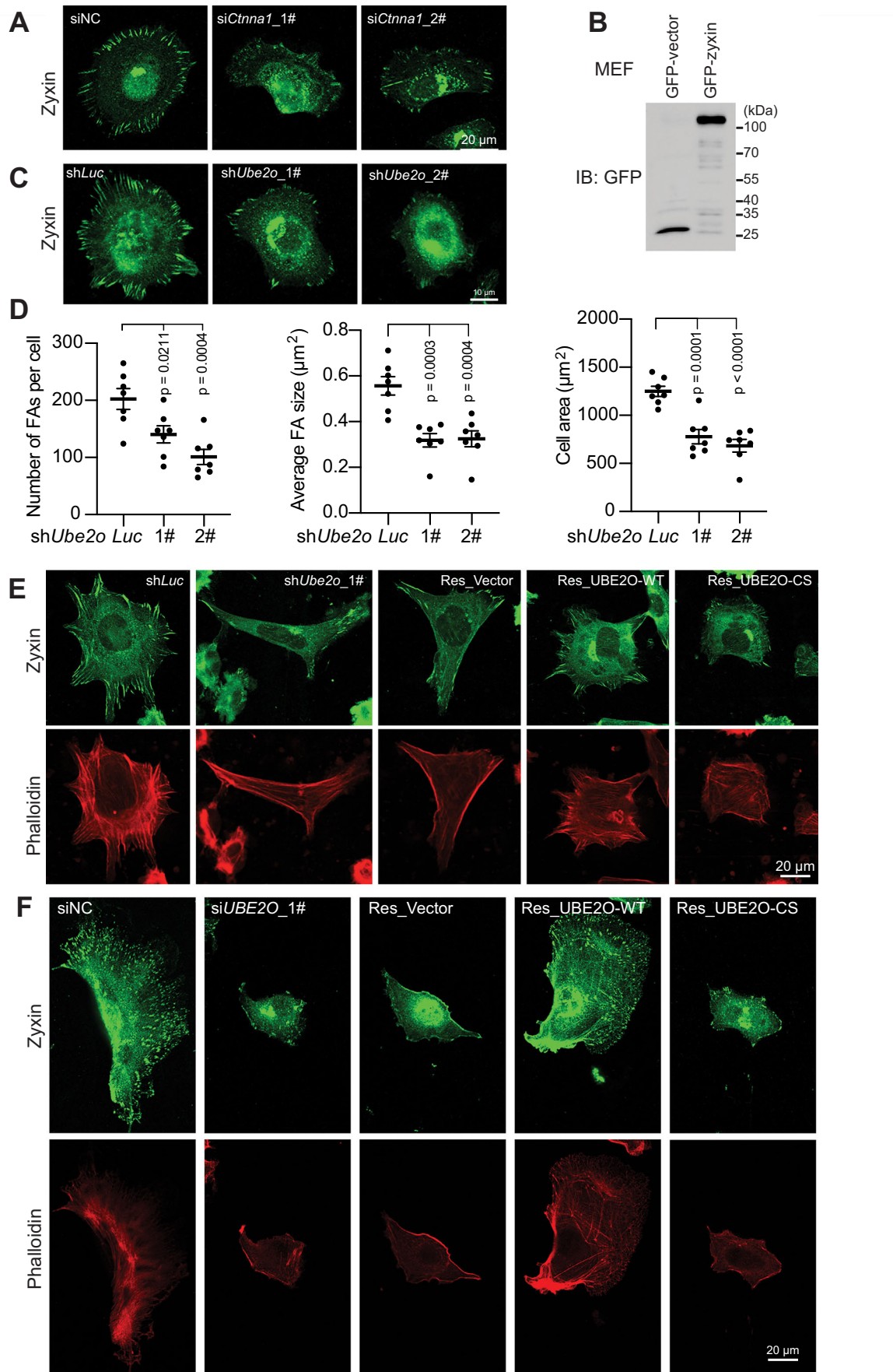

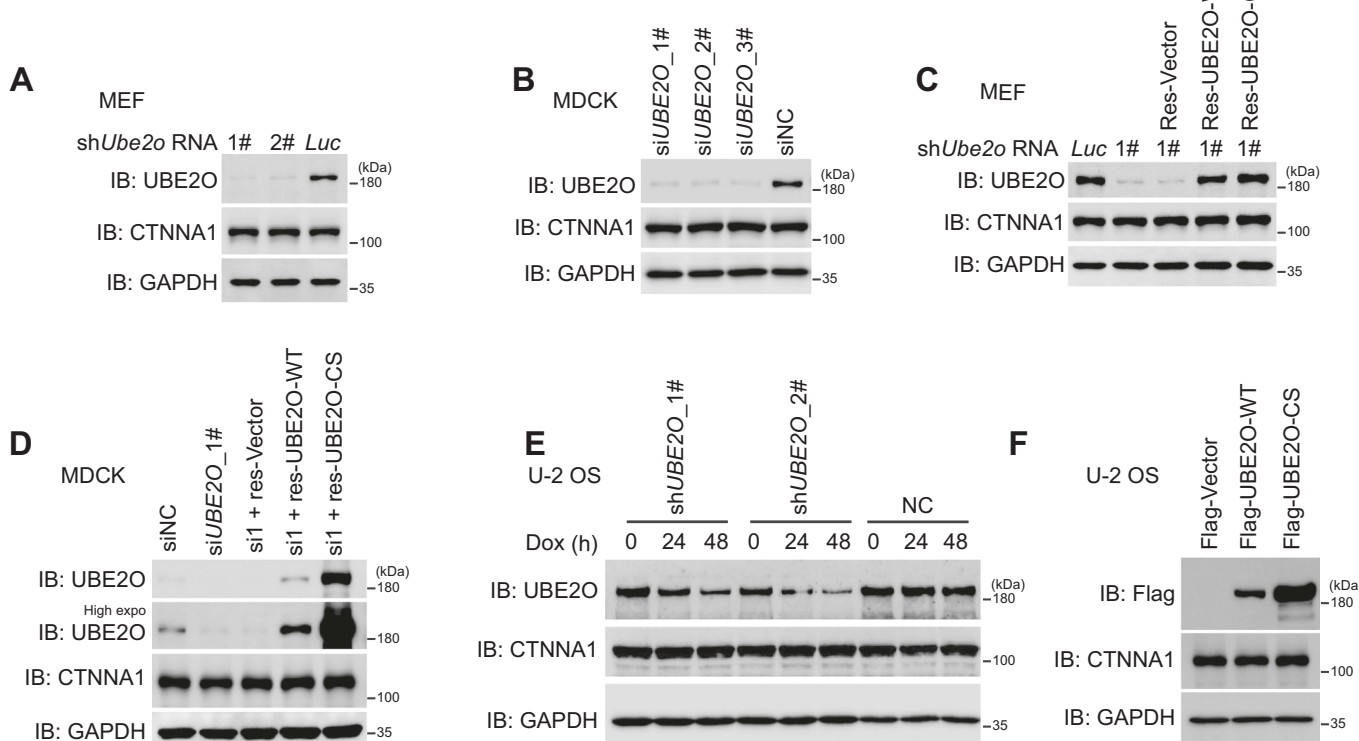

**Figure EV2. UBE2O-mediated ubiquitylaton has no effects on the stability of CTNNA1.**

(A) Knockdown of *Ube2o* in MEFs results in no significant change in CTNNA1 protein level. MEFs were treated with 500 ng/mL doxycycline for 48 h to express shRNAs targeting *Ube2o* UTR region (1#) and CDS region (2#) and a negative control targeting luciferase (sh*Luc*). Cells were harvested for western blot analyses. (B) Knockdown *UBE2O* expression in MDCK cells results in no significant change in CTNNA1 protein level. MDCK cells transfected with siNC and si*UBE2O* RNAs for 48 h were harvested for western blot analyses. (C, D) Re-introducing UBE2O into UBE2O-deficient cells had no considerable effect on CTNNA1 protein level. (C) *Ube2o* knockdown MEFs with or without rescuing the expression of UBE2O-WT, UBE2O-CS or a control vector were harvested for western blot analyses. (D) MDCK cells with or without stably expressing UBE2O-WT, UBE2O-CS mutant and a control vector were transfected with siNC and si*UBE2O* RNAs as indicated for 48 h. Cells were harvested for western blot analyses. (E) Gradual knockdown of *UBE2O* in U-2 OS cells led to no significant regulation of CTNNA1 protein level. U-2 OS cells expressing inducible shRNAs targeting *UBE2O* and a negative control (NC) were treated with 500 ng/mL doxycycline (Dox) for 0 h, 24 h and 48 h to achieve gradual knockdown of *UBE2O*. Cells were harvested for western blot analyses. (F) Ectopic expression of UBE2O in U-2 OS cells results in no notable change in CTNNA1 protein level. U-2 OS cells ectopically expressing Flag-UBE2O-WT, Flag-UBE2O-CS mutant, or Flag-vector were harvested for western blot analyses. An anti-CTNNA1 antibody was used to detect endogenous CTNNA1 for all experiments. All experiments were repeated at least twice, one representative result is shown.

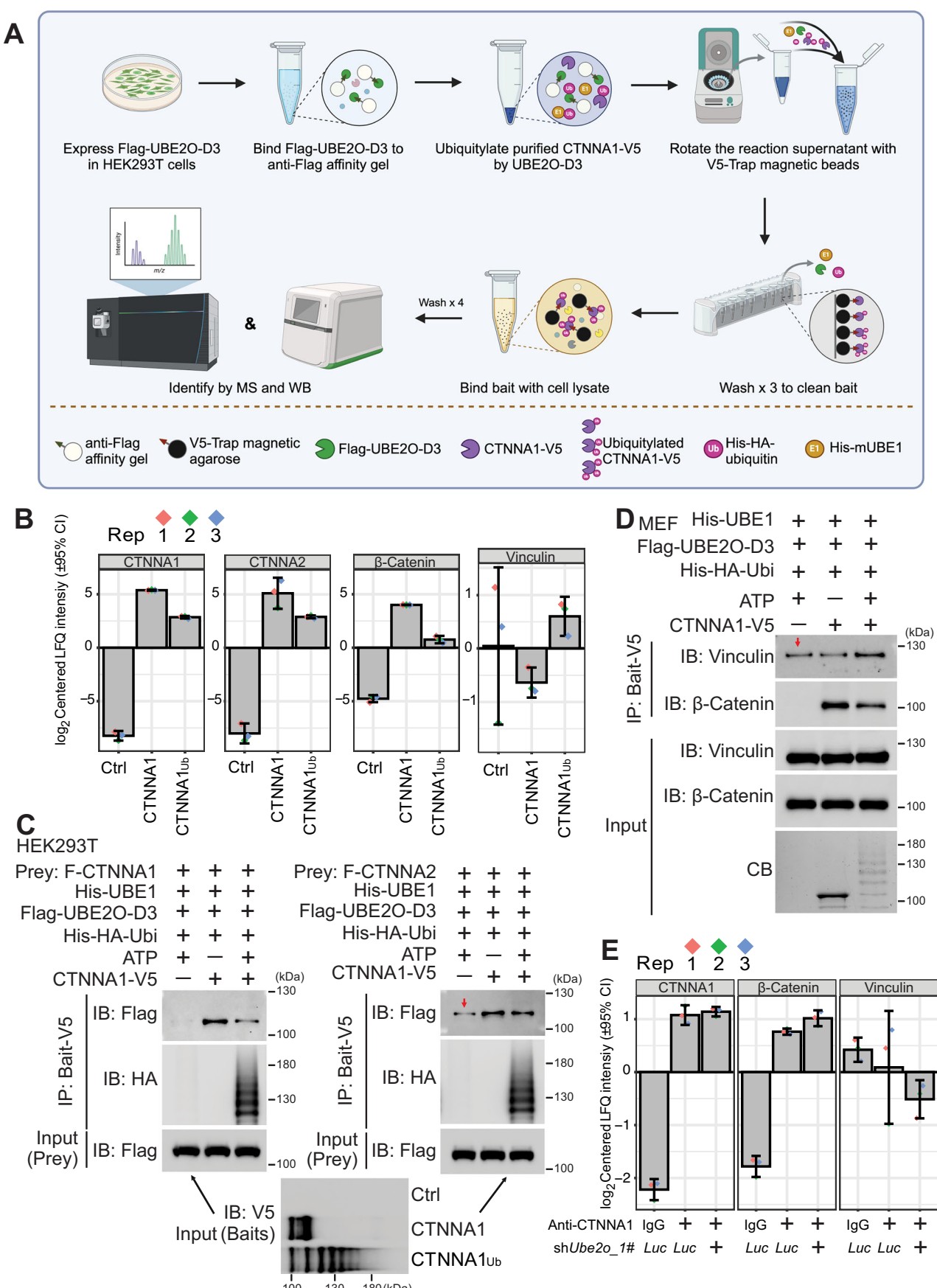

◀ **Figure EV3. UBE2O-mediated ubiquitylation of CTNNA1 alters its interactome.**

(A) Schematic diagram showing the procedure to identify the interactome of ubiquitylated CTNNA1. Detailed description is shown in the method. This diagram was created in BioRender. (B) Barplot showing the $\log_2$ Centered LFQ (label free quantification) intensity ( ±95% CI) of CTNNA1, CTNNA2, β-catenin and vinculin identified in ubiquitylated CTNNA1 interactome. Data were visualized using the DEP2 package, $n = 3$ technical replicates in independent experiments, error bars indicate mean ± SD. (C) Validation of the interaction between ubiquitylated CTNNA1 with itself or CTNNA2 by immunoprecipitation coupled with immunoblotting experiments in HEK293T cells. The ctrl, CTNNA1 and ubiquitylated CTNNA1 (CTNNA1$_{Ub}$) baits were aliquoted into 2 tubes equally and incubated with cell lysate of HEK293T cells transiently expressing Flag-CTNNA1 or Flag-CTNNA2 overnight. The proteins bound to beads were eluted and identified by immunoblotting using an anti-Flag antibody. An anti-V5 antibody was used to confirm the successful reconstitution of the baits. The red arrow indicates unspecific binding to the beads. (D) Validation of the interaction between ubiquitylated CTNNA1 and vinculin or β-catenin by immunoprecipitation coupled with immunoblotting experiments in MEFs. The ctrl, CTNNA1 and ubiquitylated CTNNA1 baits were incubated with MEF cell lysate overnight, the bound proteins were eluted and identified by immunoblotting using anti-vinculin and anti-β-catenin antibodies. Coomassie blue staining (CB) was used to confirm the successful reconstitution of the baits. The band indicated by the red arrow indicates unspecific binding to magnetic beads. (E) Barplot showing the $\log_2$ Centered LFQ (label free quantification) intensity ( ± 95% CI) of CTNNA1, β-catenin and vinculin identified in endogenous CTNNA1 interactome upon *Ube2o* knockdown. Data were visualized using the DEP2 package, $n = 3$ technical replicates in independent experiments, error bars indicate mean ± SD. Experiments were repeated at least twice, one representative result is shown.

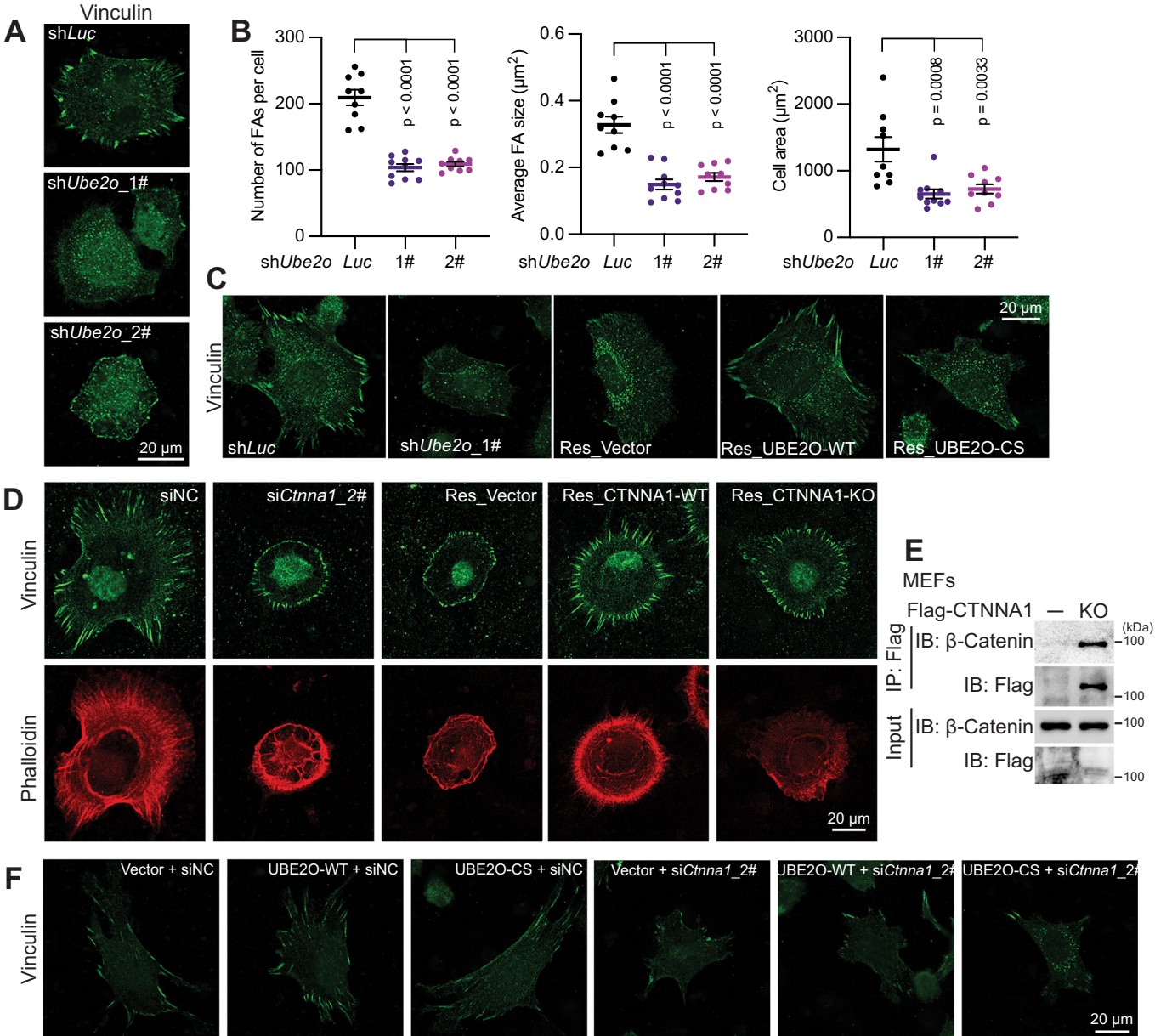

**Figure EV4. UBE2O-mediated monoubiquitylation of CTNNA1 is required for vinculin-containing FAs maturation and cell extension.**

(A, B) Knockdown of *Ube2o* impedes vinculin-containing FAs maturation. (A) Representative micrographs of MEFs treated with doxycycline for 48 h to express shRNAs targeting *Ube2o* UTR region (1#) or CDS region (2#) and a negative control targeting luciferase (sh*Luc*). After 6 h of spreading on FN-coated 8-cell chamber slides, cells were stained for vinculin. (B) Quantifications of the number of FAs per cell (vinculin channel), average FA size (vinculin channel) and cell area (vinculin channel) of the *Ube2o* knockdown MEFs 6 h after plating on FN-coated 8-cell chamber slides. Total number of cells analyzed: sh*Luc* (9), sh1 (10), sh2 (9). Error bars indicate mean ± SEM. Significance among multiple groups was determined using ANOVA followed by Tukey's post hoc test. (C) The catalytic activity of UBE2O is responsible for vinculin-containing FAs maturation. Representative micrographs of *Ube2o* knockdown MEFs with or without rescuing the expression of UBE2O-WT, UBE2O-CS mutant, or a control vector. All slides were stained for vinculin after 6 h of spreading on FN-coated 8-cell chamber slides. (D) Ubiquitylation of CTNNA1 is engaged in vinculin-containing FAs maturation and cell extension. Representative micrographs of *Ctnna1* knockdown MEFs with or without rescuing the expression of CTNNA1-WT, CTNNA1-KO mutant or a control vector. All slides were stained for vinculin and phalloidin after 6 h of spreading on FN-coated 8-cell chamber slides. (E) Immunoblot shows stably expressed Flag-CTNNA1-KO interacts with endogenous β-catenin in MEFs. (F) UBE2O regulates vinculin-containing FAs maturation via CTNNA1. Representative micrographs of MEFs expressing UBE2O-WT, UBE2O-CS mutant or a control vector transfected with a siRNA targeting *Ctnna1* or a negative control (siNC) for 48 h. All slides were stained for vinculin after 6 h of spreading on FN-coated 8-cell chamber slides. All experiments were repeated at least twice, one representative result is shown.

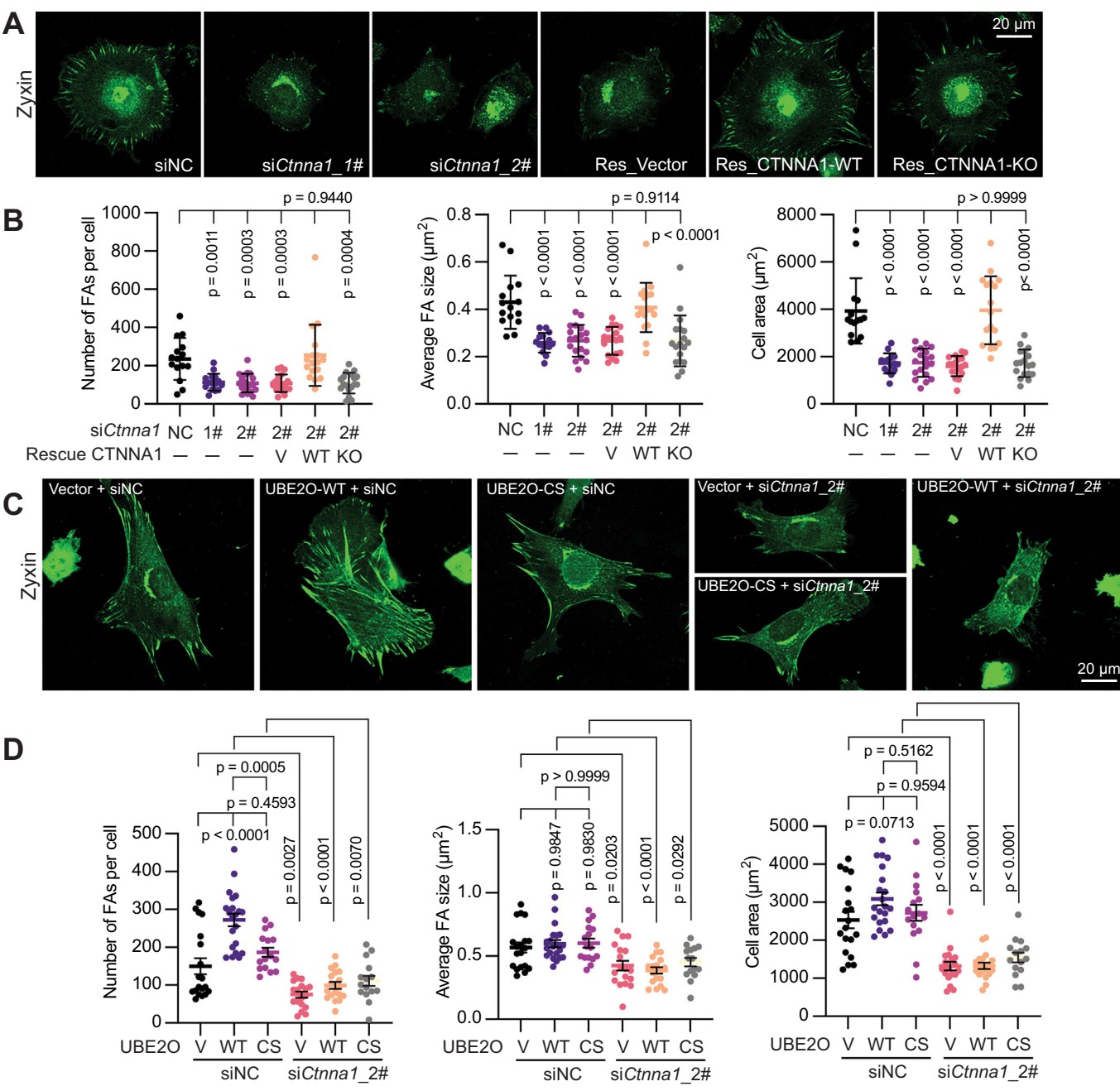

**Figure EV5. UBE2O-mediated monoubiquitylation of CTNNA1 is required for zyxin-containing FAs maturation and cell extension.**

(A, B) Ubiquitylation of CTNNA1 participates in zyxin-containing FAs maturation. (A) Representative micrographs of *Ctnna1* knockdown MEFs with or without rescuing the expression of CTNNA1-WT, CTNNA1-KO mutant or a control vector, all slides were stained for zyxin after 6 h of spreading on FN-coated 8-cell chamber slides. (B) Quantifications of the number of FAs per cell (zyxin channel), average FA size (zyxin channel) and cell area (zyxin channel) of the indicated MEFs 6 h after plating on FN-coated 8-cell chamber slides. Total number of cells analyzed: siNC (15), si*Ctnna1*_1# (15), si*Ctnna1*_2# (19), Res_Vector (18), Res_CTNNA1-WT (17), Res_CTNNA1-KO (18). (C, D) UBE2O regulates zyxin-containing FAs maturation via CTNNA1. (C) Representative micrographs of MEFs expressing UBE2O-WT, UBE2O-CS mutant or a control vector transfected with a siRNA targeting *Ctnna1* or a negative control (siNC) for 48 h, all slides were stained for zyxin after 6 h of spreading on FN-coated 8-cell chamber slides. (D) Quantifications of the number of FAs per cell (zyxin channel), average FA size (zyxin channel) and cell area (zyxin channel) of the indicated MEFs 6 h after plating on FN-coated 8-cell chamber slides. Total number of cells analyzed: Vector + siNC (19), UBE2O-WT + siNC (21), UBE2O-CS + siNC (16), Vector + si*Ctnna1*_2# (18), UBE2O-WT + si*Ctnna1*_2# (18), UBE2O-CS + si*Ctnna1*_2# (15). Error bars indicate mean ± SEM. Significance among multiple groups was determined using ANOVA followed by Tukey's post hoc test. All experiments were repeated at least twice, one representative result is shown.

