## [Peer Review File · EMBO Reports]

UBE2O-mediated ubiquitylation directs cytoplasmic CTNNA1 to promote cell-to-ECM adhesions

Dan Xiang, Wenfeng Wu, Ruona Shi, Xiaoxiao Tang, and Xiaofei Zhang

Corresponding author(s): Xiaofei Zhang (zhang_xiaofei@gibh.ac.cn)

Review Timeline:

Submission Date:	29th Apr 25
Editorial Decision:	6th Jun 25
Revision Received:	6th Aug 25
Editorial Decision:	29th Aug 25
Revision Received:	1st Sep 25
Accepted:	5th Sep 25

Editor: Deniz Senyilmaz Tiebe / Martina Rembold

Transaction Report:

Dear Dr. Zhang,

Thank you for submitting your manuscript to EMBO Reports, which was now seen by two referees, whose reports are copied below.

Referees express interest in the proposed role of CTNNA1 ubiquitination by UBE2O in regulation of cell adhesion to ECM. However, they also raise some concerns that need to be addressed to consider publication here.

I find the reports informed and constructive, and believe that addressing the concerns raised will significantly strengthen the manuscript. As the reports are below, and I think all points need to be addressed, I will not detail them here. Please contact me if you have questions or comments regarding the revision for further discussion (also by video chat).

Given these positive recommendations, we would like to invite you to revise your manuscript with the understanding that the referee concerns (as in their reports) must be fully addressed and their suggestions taken on board. Please address all referee concerns in a complete point-by-point response. Acceptance of the manuscript will depend on a positive outcome of a second round of review. It is EMBO reports policy to allow a single round of major experimental revision only and acceptance or rejection of the manuscript will therefore depend on the completeness of your responses included in the next, final version of the manuscript.

We realize that it is difficult to revise to a specific deadline. In the interest of protecting the conceptual advance provided by the work, we recommend a revision within 3 months. Please discuss the revision progress ahead of this time with me if you require more time to complete the revisions, or if you have questions or comments regarding the revision (also by video chat).

1. A data availability section providing access to data deposited in public databases is missing (where applicable).
2. Your manuscript contains statistics and error bars based on $n=2$. Please use scatter plots in these cases.

You can submit the revision either as a Scientific Report or as a Research Article. For Scientific Reports, the revised manuscript can contain up to 5 main figures and 5 Expanded View figures, and it should not exceed 27000 characters. If the revision leads to a manuscript with more than 5 main figures it will be published as a Research Article. In this case the Results and Discussion section should be separate. If a Scientific Report is submitted, these sections have to be combined. This will help to shorten the manuscript text by eliminating some redundancy that is inevitable when discussing the same experiments twice. In either case, all materials and methods should be included in the main manuscript file.

4) a .docx formatted letter INCLUDING the reviewers' reports and your detailed point-by-point responses to their comments. As part of the EMBO publication's Transparent Editorial Process, EMBO reports publishes online a Review Process File (RPF) to accompany accepted manuscripts. This File will be published in conjunction with your paper and will include the referee reports,

your point-by-point response and all pertinent correspondence relating to the manuscript.

<https://www.embopress.org/page/journal/14693178/authorguide#transparentprocess>

5) a complete author checklist, which you can download from our author guidelines

<https://www.embopress.org/page/journal/14693178/authorguide>. Please insert information in the checklist that is also reflected in the manuscript. The completed author checklist will also be part of the RPF.

6) Please note that all corresponding authors are required to supply an ORCID ID for their name upon submission of a revised manuscript (<<https://orcid.org/>>). Please find instructions on how to link your ORCID ID to your account in our manuscript tracking system in our Author guidelines

<<https://www.embopress.org/page/journal/14693178/authorguide#authorshipguidelines>>

7) Before submitting your revision, primary datasets produced in this study need to be deposited in an appropriate public database (see <https://www.embopress.org/page/journal/14693178/authorguide#datadeposition>). Please remember to provide a reviewer password if the datasets are not yet public. The accession numbers and database should be listed in a formal "Data Availability" section placed after Materials & Method (see also

<https://www.embopress.org/page/journal/14693178/authorguide#datadeposition>). Please note that the Data Availability Section is restricted to new primary data that are part of this study. * Note - All links should resolve to a page where the data can be accessed. *

Additional information on source data and instruction on how to label the files are available:

<https://www.embopress.org/page/journal/14693178/authorguide#sourcedata>

9) Our journal encourages inclusion of *data citations in the reference list* to directly cite datasets that were re-used and obtained from public databases. Data citations in the article text are distinct from normal bibliographical citations and should directly link to the database records from which the data can be accessed. In the main text, data citations are formatted as follows: "Data ref: Smith et al, 2001" or "Data ref: NCBI Sequence Read Archive PRJNA342805, 2017". In the Reference list, data citations must be labeled with "[DATASET]". A data reference must provide the database name, accession number/identifiers and a resolvable link to the landing page from which the data can be accessed at the end of the reference. Further instructions are available at <http://www.embopress.org/page/journal/14693178/authorguide#referencesformat>

10) Regarding data quantification (see Figure Legends:

<https://www.embopress.org/page/journal/14693178/authorguide#figureformat>)

12) Please also note our reference format:

13) All Materials and Methods need to be described in the main text using our 'Structured Methods' format, which is required for all research articles. According to this format, the Methods section includes a Reagents and Tools Table (listing key reagents, experimental models, software and relevant equipment and including their sources and relevant identifiers) followed by a Methods and Protocols section describing the methods using a step-by-step protocol format. The aim is to facilitate adoption of the methodologies across labs. More information on how to adhere to this format as well as a downloadable template (.docx) for the Reagents and Tools Table can be found in our author guidelines:

I look forward to seeing a revised version of your manuscript when it is ready. Please let me know if you have questions or comments regarding the revision.

Kind regards,

Deniz Senyilmaz Tiebe

Deniz Senyilmaz Tiebe, PhD
Senior Scientific Editor
EMBO Reports

Referee #1:

In this manuscript, the authors identify UBE2O as a regulator that specifically interacts with and ubiquitylates cytoplasmic α E-catenin. They show that this ubiquitination is independent of α -catenin's phosphorylation state and that it reduces α -catenin's association with β -catenin while enhancing its interaction with vinculin. This shift in molecular interactions supports the maturation of focal adhesions and promotes cell spreading.

This is a very interesting and thorough study with novel results relevant for the cell biology crowd. I have only one major comment and a few minor ones.

Major comment:

Fig. 6D and Fig. S9C - the idea that ubiquitinated a-catenin plays a role in FA regulation by enhancing its interaction with vinculin is intriguing, but also raises an important question that is not directly addressed by the authors: what happens in cases where a-catenin is recruited to AJs and interacts there with vinculin? The authors should address this.

The authors should also quantify the adhesions from the experiment of which an example is shown in Fig. 6D.

Minor comments:

Fig. 1A -- Although a-catenin is known to bind to the E-cadherin-b-catenin complex, it does not bind directly to E-cadherin. Therefore, it is likely that the proteins that were immunoprecipitated included those that were in complexes and not only direct interactors.

For readability, it would be good to provide a short explanation of the "alternative ubiquitylation theory" in the results section (Page 6) and not only in the Discussion.

Quantification of FAs - in terms of definition FAs are mature adhesions, typically elongated. While the images shown in Figs. S4 and S8 indicate the presence of mature FAs, the quantification of hundreds of FAs seems excessive. It appears that this quantification includes mature FAs as well nascent adhesions / focal complexes. Since a-catenin is a regulator of adhesion maturation, the quantification should differentiate between mature and small adhesions. Likewise for the quantification of adhesion sizes.

Fig. 7A,B - The Y-axis is "Adhesion rate (100%)". Does this mean that only between 0.3% and 0.4% of the cells adhered? Or should this be 30-40%?

Referee #2:

Manuscript ID EMBOR-2025-61827V1

Title: UBE2O-mediated ubiquitylation directs cytoplasmic CTNNA1 to promote cell-to-ECM adhesions

Xiang et al. report a novel function of CTNNA1 ubiquitylation in promoting focal adhesion. The authors employ a comprehensive set of biochemical assays, both in vivo and in vitro, to investigate the molecular basis and functional consequences of CTNNA1 multi-monoubiquitylation. They propose that this post-translational modification, mediated by the enzyme UBE2O, regulates the CTNNA1 interactome and enables a mutually exclusive interaction of unmodified CTNNA1 with adherens junction components, and CTNNA1-Ub with focal adhesion components. This newly uncovered role of CTNNA1 multi-monoubiquitylation in coordinating cell-cell and cell-ECM interactions represents a valuable contribution to our understanding of how ubiquitin regulates cellular adhesion dynamics.

I strongly recommend the publication of this work after a minor revision.

While the manuscript is already of high quality, I offer the following comments that may help strengthen the authors' message:

Major Comment

1. Although the authors use a range of approaches to study CTNNA1 multi-monoubiquitylation, most of the experiments rely on protein overexpression and/or transient transfection. It would significantly enhance the manuscript to demonstrate endogenous CTNNA1 multi-monoubiquitylation and the role of endogenous UBE2O in this process. For example, analysis of endogenous CTNNA1 ubiquitylation in control and UBE2O knockdown cells would be informative. I acknowledge the authors' remarks about experimental limitations (page 14, lines 10-12) and the potential requirement of specific stimuli for CTNNA1 ubiquitylation. I also appreciate the attempt to address this using overexpression of the UBE2O-D3 truncated mutant (Figure 2D). However, given the central role of CTNNA1-Ub in the study, it is important to provide evidence that it exists endogenously.

Minor Comments

1. The authors use the CTNNA1-KO mutant in functional assays (Supplementary Figures 7 and 8). Since the readout is negative (i.e., no rescue), it is important to verify that the CTNNA1-KO protein is functionally competent and not misfolded. A simple co-immunoprecipitation showing that CTNNA1-KO retains the ability to bind known partners (e.g., β -catenin) would address this concern.

2. The authors propose that CTNNA1 multi-monoubiquitylation mediates mutually exclusive interactions with vinculin and β -catenin. Additional experiments to test endogenous interactions would strengthen this conclusion. For example, the authors could assess endogenous CTNNA1 interactions with β -catenin and vinculin in control versus UBE2O knockdown cells. Alternatively, they could compare the binding of CTNNA1-WT and CTNNA1-KO to these partners in transient transfection experiments to determine if the interaction profile changes under CTNNA1 ubiquitylation-deficient conditions.

Response to referees for manuscript ID EMBOR-2025-61827V1

We sincerely thank both reviewers for their thoughtful and thorough evaluation of our work. We have addressed all the points raised by the reviewers and made the necessary revisions accordingly. We believe that, thanks to these comments, the revised manuscript is now stronger and more balanced. Previous Supplementary materials have been reclassified as Main/Extended View/Appendix materials as follows:

Original Fig. S3 → Revised Fig. 3;

Original Figs. S4-S8 → Revised Fig. EV1-EV5;

Original Figs. S1-S2 → Revised Appendix Fig. S1-S2;

Original Figs. S9-S10 → Revised Appendix Fig. S3-S4;

Original Dataset S1 → Revised Dataset EV1;

Original Table S1 → Revised Dataset EV2.

Additional revisions include:

Formatting adjustments to meet the journal guidelines;

New experimental data/explanations added in response to reviewers' comments (see detailed point-by-point responses).

Major textual changes in the manuscript have been highlighted in red. Below are our point-by-point responses to each of the reviewers' comments.

Referee #1:

In this manuscript, the authors identify UBE2O as a regulator that specifically interacts with and ubiquitylates cytoplasmic α -catenin. They show that this ubiquitination is independent of α -catenin's phosphorylation state and that it reduces α -catenin's association with β -catenin while enhancing its interaction with vinculin. This shift in molecular interactions supports the maturation of focal adhesions and promotes cell spreading. This is a very interesting and thorough study with novel results relevant for the cell biology crowd. I have only one major comment and a few minor ones.

Major comment:

Fig. 6D and Fig. S9C - the idea that ubiquitinated α -catenin plays a role in FA regulation by enhancing its interaction with vinculin is intriguing, but also raises an important question that is not directly addressed by the authors: what happens in cases where α -catenin is recruited to AJs and interacts there with vinculin? The authors should address this.

Major comment 1.1: What happens in cases where α -catenin is recruited to AJs and interacts there with vinculin? *The authors should address this.*

Authors' response: We appreciate the reviewer for raising this important and insightful question regarding the differential roles of α -catenin in cell-to-cell adhesions versus cell-to-ECM adhesions. The mechanosensitive interaction between α -catenin and vinculin at AJs has been well established in previous studies (Buckley *et al*, 2014; Ishiyama *et al*, 2013; Thomas *et al*, 2013; Yao *et al*, 2014; Yonemura *et al*, 2010). When incorporated into the E-cadherin/ β -catenin complex at AJs, mechanical forces induce unfolding of the autoinhibited M-domain of α -catenin, thereby exposing vinculin-binding sites. This binding forms a mechanoresponsive linkage between cadherins and the actin cytoskeleton, creating a self-reinforcing system that strengthens cell-to-cell adhesion. Importantly, phosphorylation of α -catenin at AJs enhances junctional stability without affecting its interaction with vinculin (Escobar *et al*, 2015). In our study, we identified a distinct mechanism that operates when α -catenin is not incorporated into AJ complexes. We found that UBE2O selectively interacts with and ubiquitylates cytosolic CTNNA1 (rather than the β -catenin-bound complex). This modification promotes the preferential interaction with vinculin over β -catenin, thereby leading to its recruitment to adhesion structures rather than AJs. The absence of crosstalk between the phosphorylation and ubiquitylation of CTNNA1 further supports the notion that ubiquitylated CTNNA1 does not participate in cell-to-cell adhesions. This represents a novel ubiquitin-dependent pathway that complements the known mechanosensitive pathway at AJs. We fully acknowledge that we should have better addressed this distinction in our original manuscript. We have now revised the Discussion section (Page 14, Line 27 to Page 15, Line 7) to better compare and highlight how these two distinct regulatory mechanisms allow α -catenin to function in different cellular contexts.

The authors should also quantify the adhesions from the experiment of which an example is shown in Fig. 6D.

Major comment 1.2: The authors should also quantify the adhesions from the experiment of which an example is shown in Fig. 6D.

Authors' response: We appreciate the reviewer for this constructive suggestion. We have now performed quantitative analysis of the adhesion structures from the experiment originally presented in Fig. 6D, which is included as Figure 7D in the revised manuscript.

Minor comments:

Fig. 1A -- Although α -catenin is known to bind to the E-cadherin- β -catenin complex, it does not bind directly to E-cadherin. Therefore, it is likely that the proteins that were immunoprecipitated included those that were in complexes and not only direct interactors.

Minor comment 1: it is likely that the proteins that were immunoprecipitated included those that were in complexes and not only direct interactors.

Authors' response: We sincerely appreciate the reviewer's insightful comment regarding the composition of immunoprecipitated protein complexes. The reviewer's comment is correct: under the experimental conditions described in Fig. 1A, co-immunoprecipitated proteins may include both direct interactors and components of larger protein complexes. To specifically address whether UBE2O interacts directly with α -catenin, we conducted additional *in vitro* binding assays using purified proteins. As shown in the original Fig. 1D and E, these controlled *in vitro* experiments, free of contaminants from potential cellular complexes, demonstrate that UBE2O and α -catenin can interact directly, without the need for additional cellular factors. We have now emphasized this point more clearly in the Results section (Page 5, Lines 19-20) of the revised manuscript.

For readability, it would be good to provide a short explanation of the "alternative ubiquitylation theory" in the results section (Page 6) and not only in the Discussion.

Minor comment 2: For readability, it would be good to provide a short explanation of the "alternative ubiquitylation theory" in the results section (Page 6) and not only in the Discussion.

Authors' response: We thank the reviewer for this helpful suggestion to enhance the manuscript's clarity. We have now added a concise explanation of the "alternative ubiquitylation theory" in the Results section (Page 7, Lines 12-15). In addition, in order to avoid redundancy, we have also revised the corresponding paragraph in the Discussion (Page 15, Lines 24-26).

Quantification of FAs - in terms of definition FAs are mature adhesions, typically elongated. While the images shown in Figs. S4 and S8 indicate the presence of mature FAs, the quantification of hundreds of FAs seems excessive. It appears that this quantification includes mature FAs as well nascent adhesions / focal complexes. Since a-catenin is a regulator of adhesion maturation, the quantification should differentiate between mature and small adhesions. Likewise for the quantification of adhesion sizes.

Minor comment 3: Since a-catenin is a regulator of adhesion maturation, the quantification should differentiate between mature and small adhesions. Likewise for the quantification of adhesion sizes.

Authors' response:

We appreciate the reviewer's insightful critique regarding FAs quantification and acknowledge the need for precise terminology. Specifically, our analysis included all detectable focal adhesion structures positive for zyxin, following careful threshold adjustment to minimize background signal. We analyze the data for the following considerations:

1) The biological specificity of zyxin excludes nascent adhesions. Current consensus (Kanchanawong *et al*, 2010; Zaidel-Bar *et al*, 2003) definitively establishes zyxin as an FA marker absent in nascent adhesions and focal complexes.

2) UBE2O/CTNNA1 function initiates early in adhesion maturation. As demonstrated in Fig. 4A, UBE2O and CTNNA1 co-localize on small FAs (0.5-1 μm), with intensity scaling progressively through the FA maturation stages. This evidence necessitates the inclusion of small FAs to capture the functional role of UBE2O/CTNNA1 in early adhesion maturation stages.

3) Holistic quantification provides more information beyond sole mature FAs assessment. We validated our approach against the Fiji focal adhesion analysis plugin (Pan *et al*, 2020; Sun *et al*, 2025). The number of mature FAs analysed by this plugin (Fig. R1A) is a evidence for adhesion maturation, while this is visually apparent in representative micrographs in original Fig. S4, S7 and S8. The changes in the number and size distribution of all zyxin-positive FAs can also provide a robust and physiologically relevant readout of the overall efficiency of adhesion maturation (Fig. R1B, or the original Fig. S4D). Specifically: an increase in the number of FAs indicates enhanced adhesion stabilization (preventing rapid disassembly of nascent adhesions), a shift towards larger adhesion sizes directly reflects progression through maturation stages (Chastney *et al*, 2021). Therefore, our statistical approach captures more information about the number and size distribution shifts that reflect the progression dynamics. Based on the above considerations, we retained manual thresholding to comprehensively cover the functional maturation continuum (small FAs, mature FAs and fibrillar adhesions).

Figure R1 A, focal adhesion analysis Fiji plugin quantified number of mature FAs per cell (zyxin channel) and average mature FA size (zyxin channel) of the indicated MEFs 6 h after plating on FN-coated 8-cell chamber slides in original Fig. S4C and D. **B**, original Fig. S4D shows the quantification of number of mature FAs per cell (zyxin channel) and average mature FA size (zyxin channel) of the *Ube2o* knockdown MEFs 6 h after plating on FN-coated 8-cell chamber slides using manual thresholding. $n = 7$ cells in all cases, all quantifications represent the mean \pm SEM and comparisons were performed by ANOVA followed by Tukey's post hoc test.

To accurately reflect the biology we observed and address the reviewer's valid point, we have therefore implemented the following revisions: 1) We have strengthened the results statements regarding the localization of UBE2O/CTNNA1 at FAs (Page 9, Lines 1-2). 2) Update the text to consistently use "FAs maturation" when referring to the related micrographs and quantifications. We believe this revision improves the clarity and accuracy of our data presentation and interpretation.

Fig. 7A,B - The Y-axis is "Adhesion rate (100%)". Does this mean that only between 0.3% and 0.4% of the cells adhered? Or should this be 30-40%?

Minor comment 4: Fig. 7A,B - The Y-axis is "Adhesion rate (100%)". Does this mean that only between 0.3% and 0.4% of the cells adhered? Or should this be 30-40%?

Authors' response: The Y-axis indicates that 30-40% of the cells adhered. We sincerely appreciate the reviewer's careful attention to this detail in our adhesion assay data. We acknowledge that the Y-axis labeling in the original Fig. 7A and B was unclear and could lead to misinterpretation. We have now revised Y-axis to read "Adhesion rate (%)" with appropriate scale adjustments (Fig. 8A, B in revised manuscript).

Referee #2:

Manuscript ID EMBOR-2025-61827V1

Title: UBE2O-mediated ubiquitylation directs cytoplasmic CTNNA1 to promote cell-to-ECM adhesions

Xiang et al. report a novel function of CTNNA1 ubiquitylation in promoting focal adhesion. The authors employ a comprehensive set of biochemical assays, both in vivo and in vitro, to investigate the molecular basis and functional consequences of CTNNA1 multi-monoubiquitylation. They propose that this post-translational modification, mediated by the enzyme UBE2O, regulates the CTNNA1 interactome and enables a mutually exclusive interaction of unmodified CTNNA1 with adherens junction components, and CTNNA1-Ub with focal adhesion components. This newly uncovered role of CTNNA1 multi-monoubiquitylation in coordinating cell-cell and cell-ECM interactions represents a valuable contribution to our understanding of how ubiquitin regulates cellular adhesion dynamics.

I strongly recommend the publication of this work after a minor revision.

While the manuscript is already of high quality, I offer the following comments that may help strengthen the authors' message:

Major Comment

1. Although the authors use a range of approaches to study CTNNA1 multi-monoubiquitylation, most of the experiments rely on protein overexpression and/or transient transfection. It would significantly enhance the manuscript to demonstrate endogenous CTNNA1 multi-monoubiquitylation and the role of endogenous UBE2O in this process. For example, analysis of endogenous CTNNA1 ubiquitylation in control and UBE2O knockdown cells would be informative. I acknowledge the authors' remarks about experimental limitations (page 14, lines 10-12) and the potential requirement of specific stimuli for CTNNA1 ubiquitylation. I also appreciate the attempt to address this using overexpression of the UBE2O-D3 truncated mutant (Figure 2D). However, given the central role of CTNNA1-Ub in the study, it is important to provide evidence that it exists endogenously.

Major comment 1: given the central role of CTNNA1-Ub in the study, it is important to provide evidence that it exists endogenously.

Authors' response: We thank the reviewer for their kind words and for raising this critical point regarding the endogenous detection of ubiquitylated CTNNA1, which is indeed fundamental to our study's conclusions. Unfortunately, we encountered three major technical limitations: 1) Antibody affinity: standard immunoprecipitation conditions for ubiquitylation assay (requiring 0.1% SDS or 8M urea) are incompatible with available anti-CTNNA1 antibodies, as this antibody fails to immunoprecipitate any CTNNA1 protein (Fig. R2A). 2) Basal level sensitivity issues: even when using the ubiquitin antibody P4D1, the basal level of endogenous CTNNA1 ubiquitylation is below reliable detection thresholds (Fig. R2B). This limitation persists unless D3 is overexpressed to amplify the signal. 3) Mass spectrometry limitations: proteomic analysis of UBE2O knockdown versus control cells failed to detect endogenous CTNNA1 ubiquitylation due to its low abundance. We have now explicitly discussed these technical limitations and complementary strategies in the revised Discussion (Page 15, Lines 16-21), while emphasizing the need for future study on the stimuli for CTNNA1 ubiquitylation, development of higher-affinity CTNNA1 antibodies or ultrasensitive ubiquitylation identification proteomics methods.

Figure R2 **A**, immunoblot shows that anti-CTNNA1 antibody lost its ability in CTNNA1 enrichment in 0.1% SDS radioimmunoassay buffer. **B**, immunoblot shows no detectable

endogenously ubiquitylated CTNNA1 in UBE2O-deficient and control HEK293T cells using P4D1-IP. Experiments were repeated twice, one representative result is shown.

Minor Comments

1. The authors use the CTNNA1-KO mutant in functional assays (Supplementary Figures 7 and 8). Since the readout is negative (i.e., no rescue), it is important to verify that the CTNNA1-KO protein is functionally competent and not misfolded. A simple co-immunoprecipitation showing that CTNNA1-KO retains the ability to bind known partners (e.g., β -catenin) would address this concern.

Minor comment 1: it is important to verify that the CTNNA1-KO protein is functionally competent and not misfolded. A simple co-immunoprecipitation showing that CTNNA1-KO retains the ability to bind known partners (e.g., β -catenin) would address this concern.

Authors' response: We sincerely thank the reviewer for raising this critical point regarding the functional competence of the CTNNA1-KO mutant used in our rescue experiments (original Figs. S7/S8). We fully agree that confirming the structural integrity and binding capability of the KO mutant is essential to interpret the negative rescue phenotype correctly. To directly address the reviewer's concern, we have performed additional co-immunoprecipitation (Co-IP) experiments to assess the interaction between CTNNA1-KO and β -catenin (Fig. EV4E in revised manuscript).

1) **Validation in HEK293T cells (transient expression):** we expressed Flag-tagged CTNNA1-KO in HEK293T cells, immunoprecipitation using anti-Flag agarose successfully pulled down β -catenin alongside CTNNA1-KO (Fig. R3A); 2) **Validation in MEFs (stable expression):** immunoprecipitation using anti-Flag agarose in the stable MEF cell line expressing Flag-tagged CTNNA1-KO also robustly co-precipitated endogenous β -catenin (Fig. R3B, Fig. EV4E in revised version). These interaction assays demonstrate that the CTNNA1-KO protein is correctly folded and functionally competent in terms of maintaining this core molecular interaction.

Figure R3 A, immunoblot shows transiently expressed Flag-CTNNA1-KO interacts with endogenous β -catenin in HEK293T cells. B, immunoblot shows stably expressed Flag-CTNNA1-KO interacts with endogenous β -catenin in MEFs. Experiments were repeated twice, one representative result is shown.

2. The authors propose that CTNNA1 multi-monoubiquitylation mediates mutually exclusive interactions with vinculin and β -catenin. Additional experiments to test endogenous interactions would strengthen this conclusion. For example, the authors could assess endogenous CTNNA1 interactions with β -catenin and vinculin in control versus UBE2O knockdown cells. Alternatively, they could compare the binding of CTNNA1-WT and CTNNA1-KO to these partners in transient transfection experiments to determine if the interaction profile changes under CTNNA1 ubiquitylation-deficient conditions.

Minor comment 2: Additional experiments to test endogenous interactions would strengthen this conclusion. For example, the authors could assess endogenous CTNNA1 interactions with β -catenin and vinculin in control versus UBE2O knockdown cells. Alternatively, they could compare the binding of CTNNA1-WT and CTNNA1-KO to these partners in transient transfection experiments to determine if the interaction profile changes under CTNNA1 ubiquitylation-deficient conditions.

Authors' response: We thank the reviewer for this excellent suggestion to further validate the proposed model that CTNNA1 ubiquitylation acts as a molecular switch regulating its mutually exclusive interactions with β -catenin and vinculin. We fully agree that assessing endogenous interactions is crucial, and we have now performed both experimental approaches suggested by the reviewer, yielding consistent results that strongly support our model.

1) **Comparing Interactions of CTNNA1-WT vs. CTNNA1-KO.** As suggested, we compared the interaction profiles of Flag-tagged CTNNA1-WT and CTNNA1-KO with β -catenin and vinculin via Co-IP in transiently transfected HEK293T cells. CTNNA1-KO showed enhanced interaction with β -catenin and reduced interaction with vinculin compared to CTNNA1-WT (Fig. R4A). However, we noted that the expression level of CTNNA1-KO was generally lower than that of WT in these transient transfections. To rigorously determine if the reduced vinculin binding was specifically due to loss of ubiquitylation rather than lower input, we performed a critical control experiment using the CTNNA1-L344A mutant (a well-characterized point mutant deficient in vinculin binding (Mukherjee *et al*, 2022; Peng *et al*, 2012)). We co-expressed Flag-CTNNA1-WT or Flag-CTNNA1-L344A with or without UBE2O-D3 and Ubiquitin. Ubiquitylated CTNNA1-WT showed enhanced interaction with vinculin, while CTNNA1-L344A showed deficient vinculin interaction, despite robust ubiquitylation and comparable expression levels to WT (Fig. R4B). This confirms that the increased vinculin binding is dependent on CTNNA1 ubiquitylation.

Figure R4 A, immunoblot shows the interaction between CTNNA1-WT/CTNNA1-KO and endogenous β -catenin/vinculin in HEK293T cells. **B**, immunoblot shows the interaction between CTNNA1-WT/CTNNA1-L344A and endogenous β -catenin/vinculin in HEK293T cells, with or without UBE2O-D3 mediated ubiquitylation. LA: L344A. The band indicated by the red arrow indicates vinculin band. Experiments were repeated three times, one representative result is shown.

2) Assessing Endogenous Interactions in *Ube2o* Knockdown vs. Control Cells. As suggested by the reviewer, we directly analyzed changes in the endogenous interactome of CTNNA1 upon depletion of *Ube2o*. We performed immunoprecipitation of endogenous CTNNA1 from control (*shLuc*) and *Ube2o*-knockdown (*shUbe2o*) MEF cells, followed by mass spectrometry measurement to identify and quantify co-precipitated proteins. In *Ube2o*-knockdown cells, endogenous CTNNA1 showed enriched interaction with β -catenin while decreased binding to vinculin compared to controls (Fig. R4C, Fig. EV3E in revised version).

Figure R4 C, Barplot showing the log₂ Centered LFC (label free quantification) intensity ($\pm 95\%$ CI) of CTNNA1, β -catenin and vinculin identified in endogenous CTNNA1 interactome upon *Ube2o* deficient. Data were visualized using the DEP2 package, n=3 technical replicates in independent experiments, error bars indicate mean \pm SD.

We believe these two independent lines of evidence address the reviewer's request for additional endogenous validation of the conclusion regarding the ubiquitylation-dependent switch in CTNNA1 interactions. The new and crucial data in MEFs demonstrating the endogenous

interactome of CTNNA1 upon depletion of *Ube2o* has been added as Figure EV3E in the revised manuscript (expanded from original Fig. S6) and explained in revised Results (Page 10, Lines 23-25).

References

- Buckley CD, Tan J, Anderson KL, Hanein D, Volkmann N, Weis WI, Nelson WJ, Dunn AR (2014) Cell adhesion. The minimal cadherin-catenin complex binds to actin filaments under force. *Science* 346: 1254-1261
- Chastney MR, Conway JRW, Ivaska J (2021) Integrin adhesion complexes. *Curr Biol* 31: R536-R542
- Escobar DJ, Desai R, Ishiyama N, Folmsbee SS, Novak MN, Flozak AS, Daugherty RL, Mo R, Nanavati D, Sarpal R *et al* (2015) α -Catenin phosphorylation promotes intercellular adhesion through a dual-kinase mechanism. *J Cell Sci* 128: 1150-1165
- Ishiyama N, Tanaka N, Abe K, Yang YJ, Abbas YM, Umitsu M, Nagar B, Bueler SA, Rubinstein JL, Takeichi M *et al* (2013) An autoinhibited structure of α -catenin and its implications for vinculin recruitment to adherens junctions. *J Biol Chem* 288: 15913-15925
- Kanchanawong P, Shtengel G, Pasapera AM, Ramko EB, Davidson MW, Hess HF, Waterman CM (2010) Nanoscale architecture of integrin-based cell adhesions. *Nature* 468: 580-584
- Mukherjee A, Melamed S, Damouny-Khoury H, Amer M, Feld L, Nadjar-Boger E, Sheetz MP, Wolfenson H (2022) α -Catenin links integrin adhesions to F-actin to regulate ECM mechanosensing and rigidity dependence. *J Cell Biol* 221
- Pan M, Chew TW, Wong DCP, Xiao J, Ong HT, Chin JFL, Low BC (2020) BNIP-2 retards breast cancer cell migration by coupling microtubule-mediated GEF-H1 and RhoA activation. *Sci Adv* 6: eaaz1534
- Peng X, Maiers JL, Choudhury D, Craig SW, DeMali KA (2012) α -Catenin uses a novel mechanism to activate vinculin. *J Biol Chem* 287: 7728-7737
- Sun AR, Ramli MFH, Shen X, Kannivadi Ramakanth K, Chen D, Hu Y, Vidyasekar P, Foo RS, Long Y, Zhu J *et al* (2025) Hybrid hydrogel-extracellular matrix scaffolds identify biochemical and mechanical signatures of cardiac ageing. *Nat Mater*
- Thomas WA, Boscher C, Chu YS, Cuvelier D, Martinez-Rico C, Seddiki R, Heysch J, Ladoux B, Thiery JP, Mege RM *et al* (2013) α -Catenin and Vinculin Cooperate to Promote High E-cadherin-based Adhesion Strength. *J Biol Chem* 288: 4957-4969
- Yao MX, Qiu W, Liu RC, Efremov AK, Cong PW, Seddiki R, Payre M, Lim CT, Ladoux B, Mège RM *et al* (2014) Force-dependent conformational switch of α -catenin controls vinculin binding. *Nat Commun* 5
- Yonemura S, Wada Y, Watanabe T, Nagafuchi A, Shibata M (2010) α -Catenin as a tension transducer that induces adherens junction development. *Nat Cell Biol* 12: 533-U535
- Zaidel-Bar R, Ballestrem C, Kam Z, Geiger B (2003) Early molecular events in the assembly of matrix adhesions at the leading edge of migrating cells. *J Cell Sci* 116: 4605-4613

Dear Dr. Zhang,

Thank you for submitting your revised manuscript. It has now been seen by both of the original referees.

As you will see, referees find that the study is significantly improved during revision and recommend publication. However, the editorial points below need to be addressed before I can accept the manuscript.

- Datasets should be uploaded as individual files, instead of zipped with README files which should be included as individual tabs/sheets as Dataset legends in each Excel file.
- Please add page numbers to the Table of Contents of the Appendix file. Please submit the Appendix file as a pdf file.
- Please make the dataset PXD058956 publicly available and remove the reviewer token from the manuscript text.
- We note that there is a Declaration of AI-assisted writing, which should be placed below DAS.
- Our production/data editors have asked you to clarify several points in the figure legends - Figure Legends (main + EV):
 - o Please note that the exact p values are not provided in the legends of figures 4B, C; 6A-C; 7D, EV1 A, EV2 D, EV4 B, EV5 B, D.
 - o Please indicate the statistical test used for data analysis in the legends of figures 1A, EV2 D.
 - o Please note that information related to n is missing in the legend of figure 1A.
 - o Please note that the error bars are not defined in the legend of figure EV2 D.

Thank you again for giving us to consider your manuscript for EMBO Reports, I look forward to your minor revision.

Kind regards,

Deniz Senyilmaz Tiebe

--

Deniz Senyilmaz Tiebe, PhD
Senior Scientific Editor
EMBO Reports

Referee #1:

The authors have successfully addressed all of my comments. I recommend publication of the manuscript.

Referee #2:

Manuscript ID EMBOR-2025-61827V1

Title: UBE2O-mediated ubiquitylation directs cytoplasmic CTNNA1 to promote cell-to-ECM adhesions

I have reviewed the revised version of the manuscript. The authors have satisfactorily addressed all my comments. I recommend the manuscript for publication

Response to editors for manuscript ID EMBOR-2025-61827V3

We sincerely thank the editors and both reviewers for helping us to improve the quality and clarity of our manuscript. We have carefully addressed all the editorial points:

Datasets should be uploaded as individual files, instead of zipped with README files which should be included as individual tabs/sheets as Dataset legends in each Excel file.

Done.

- Please add page numbers to the Table of Contents of the Appendix file. Please submit the Appendix file as a pdf file.

Done.

- Please make the dataset PXD058956 publicly available and remove the reviewer token from the manuscript text.

Done.

- We note that there is a Declaration of AI-assisted writing, which should be placed below DAS.

Done.

- Our production/data editors have asked you to clarify several points in the figure legends - Figure Legends (main + EV):

- o Please note that the exact p values are not provided in the legends of figures 4B, C; 6A-C; 7D, EV1 A, EV2 D, EV4 B, EV5 B, D.

- p values are provided in the respective figures. All p values are reported as exact values unless $p < 0.0001$ or $p > 0.9999$, in which case they are reported as $p < 0.0001$ or $p > 0.9999$. We apologize for any confusion. Figure EV1 A and EV2 D are images that do not contain statistical graphs. In response to this point, we have ensured that exact p-values are now provided for all statistical graphs in the revised manuscript.

- o Please indicate the statistical test used for data analysis in the legends of figures 1A, EV2 D. To ensure completeness, we have reviewed all statistical figure legends. The statistical test for Figure 1A and EV1 D has been clearly stated in the revised legends.

- o Please note that information related to n is missing in the legend of figure 1A.

Done.

- o Please note that the error bars are not defined in the legend of figure EV2 D.

- We have reviewed all statistical figure legends. The error bars for all statistical figures have been clearly stated in their revised legends.

Dr. Xiaofei Zhang
Guangzhou Institutes of Biomedicine and Health, Chinese Academy of Sciences
Laboratory of Proteomics and Ubiquitin Signaling, Center for Cell Lineage Atlas
190 Kaiyuan Avenue, Guangzhou
Guangdong 510530
China

Dear Dr. Zhang,

My colleague Deniz is currently out of office and I have therefore taken over the handling of your manuscript. Thank you for implementing the final minor editorial requests. I am now very pleased to accept your manuscript for publication in the next available issue of EMBO reports. Thank you for your contribution to our journal.

Kind regards,
